# StreamForest: Efficient Online Video Understanding with Persistent Event Memory

**Xiangyu Zeng**[*1,2], **Kefan Qiu**[*1], **Qingyu Zhang**[*1], **Xinhao Li**[1], **Jing Wang**[1], **Jiaxin Li**[1]
**Ziang Yan**[3,2], **Kun Tian**[4], **Meng Tian**[5], **Xinhai Zhao**[4], **Yi Wang**[2], **Limin Wang**[†1,2]

[1]Nanjing University    [2]Shanghai AI Laboratory    [3]Zhejiang University
[4]Noah's Ark Lab, Huawei    [5]Yinwang Intelligent Tech.

https://github.com/MCG-NJU/StreamForest

## Abstract

Multimodal Large Language Models (MLLMs) have recently achieved remarkable progress in video understanding. However, their effectiveness in real-time streaming scenarios remains limited due to storage constraints of historical visual features and insufficient real-time spatiotemporal reasoning. To address these challenges, we propose **StreamForest**, a novel architecture specifically designed for streaming video understanding. Central to StreamForest is the Persistent Event Memory Forest, a memory mechanism that adaptively organizes video frames into multiple event-level tree structures. This process is guided by penalty functions based on temporal distance, content similarity, and merge frequency, enabling efficient long-term memory retention under limited computational resources. To enhance real-time perception, we introduce a Fine-grained Spatiotemporal Window, which captures detailed short-term visual cues to improve current scene perception. Additionally, we present **OnlineIT**, an instruction-tuning dataset tailored for streaming video tasks. OnlineIT significantly boosts MLLM performance in both real-time perception and future prediction. To evaluate generalization in practical applications, we introduce **ODV-Bench**, a new benchmark focused on real-time streaming video understanding in autonomous driving scenarios. Experimental results demonstrate that StreamForest achieves the state-of-the-art performance, with accuracies of 77.3% on StreamingBench, 60.5% on OVBench, and 55.6% on OVO-Bench. In particular, even under extreme visual token compression (limited to 1024 tokens), the model retains 96.8% of its average accuracy in eight benchmarks relative to the default setting. These results underscore the robustness, efficiency, and generalizability of StreamForest for streaming video understanding.

## 1 Introduction

In recent years, multimodal large language models have made significant progress in video understanding tasks, demonstrating strong semantic comprehension and reasoning capabilities across videos of varying durations and scenarios [30, 52, 31, 35]. Benefiting from large-scale pretraining and enhanced cross-modal modeling capabilities, these models have been widely adopted in various domains [28, 42, 55, 59]. However, with the growing demand for real-time intelligent processing in online applications such as autonomous driving [49], live video streaming [6], and robotics [79], researchers have increasingly shifted their focus from conventional offline video understanding to the more challenging task of streaming video processing [5, 23, 46].

---

[*] Equal contribution.

[†] Corresponding author.

39th Conference on Neural Information Processing Systems (NeurIPS 2025).

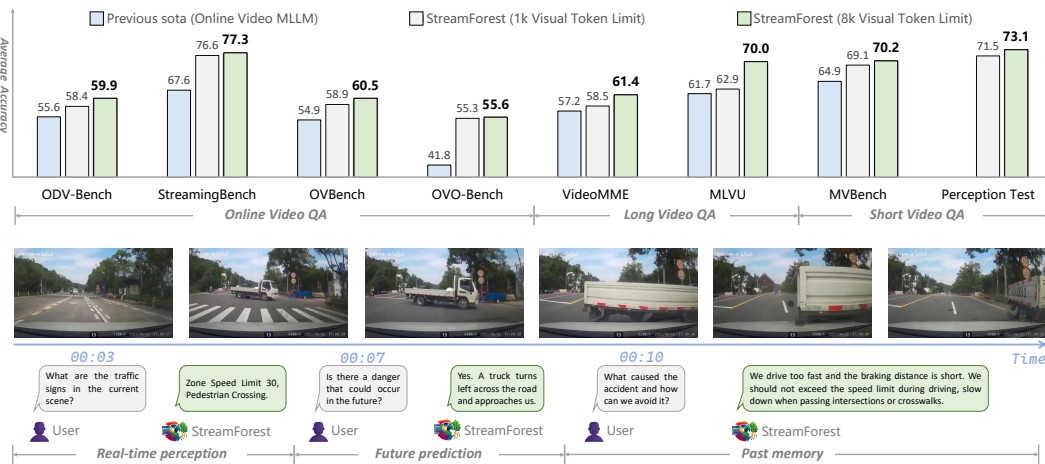

Figure 1: StreamForest achieves strong performance across various evaluation benchmarks while using significantly fewer visual tokens. It effectively handles key tasks in streaming video scenarios, including past memory, real-time perception, and future prediction.

In the field of streaming video understanding, efficiently caching continuously arriving video frame features remains a long-standing and challenging problem. To mitigate the storage and computational overhead associated with past frames, prior work has primarily employed two strategies for visual feature reduction: compression during sampling [5, 46, 64] and compression during storage [52, 72, 23]. Compression during sampling reduces a large portion of incoming visual features, which severely limits the model's capacity for fine-grained spatiotemporal reasoning. As a result, it can only perform coarse semantic summarization of the current scene. Conversely, compression during storage typically involves merging or discarding adjacent frames based on inter-frame similarity. While more memory-efficient, this strategy is susceptible to missing critical foreground actions due to background noise. It may also result in excessive local merging, introducing spatiotemporal irregularities that degrade the model's ability to retain and reason about key events over time.

To address the challenges of streaming video understanding, we propose a novel architecture called **StreamForest**. At its core is the **Persistent Event Memory Forest**, a mechanism designed to efficiently store and manage long-term visual information. This memory system enables a MLLM to process ultra-long streaming video at a constant rate of 1 fps by dynamically organizing video segments into a tree structure based on event boundaries. The merging of segments is guided by three penalty functions that consider temporal distance, content similarity, and merge frequency, ensuring an adaptive and meaningful memory hierarchy. To enhance real-time perception, we introduce a **Fine-grained Spatiotemporal Window**, which extracts rich local spatiotemporal features from nearby frames. This module enables the MLLM to better understand the current scene by focusing on temporally relevant visual context. We also present **OnlineIT**, a fine-tuning dataset specifically designed for streaming video understanding. OnlineIT improves the MLLM's ability to perceive the present moment and anticipate future events by leveraging both recent observations and long-term historical cues. It addresses the problem of hallucinations caused by spatiotemporal distribution shifts between past and current frames. In addition, we introduce **ODV-Bench**, a new benchmark for evaluating streaming video understanding in autonomous driving scenarios. ODV-Bench emphasizes real-time perception and future prediction, providing a systematic framework for assessing the generalization and real-world effectiveness of streaming video MLLMs in downstream tasks.

We conducted extensive experiments on both online and offline video understanding benchmarks to validate the effectiveness of StreamForest. Under the default setting with a visual token limit of 8192, StreamForest significantly outperforms previous state-of-the-art streaming video understanding MLLMs. It achieves an average accuracy of 77.3% on StreamingBench, 60.5% on OVBench, and 55.6% on OVO-Bench. StreamForest also matches or surpasses the performance of leading offline video understanding MLLMs on both long and short video benchmarks, despite operating in a streaming video input setting. Moreover, StreamForest demonstrates strong resilience under extreme compression. With a reduced visual token limit of just 1024, it retains 96.8% of its average performance across eight benchmarks compared to the default setting. These results highlight the robustness and efficiency of our approach in continuously processing streaming video input.

## 2 Related work

**Multimodal Large Language Model.**   Extending multimodal capabilities from static images to dynamic video sequences introduces additional complexity, requiring models to possess stronger abilities in modeling long-range dependencies and understanding events [30, 40, 81, 41, 71, 58]. Recent advances in MLLMs for video have introduced a variety of innovative strategies to tackle the challenges of efficiently processing and reasoning over long video inputs [37, 42, 29, 51, 33]. LongVILA [7] proposes a Multimodal Sequence Parallelism system for long-context modeling, enabling efficient parallel training and inference on extended video content. However, most current research on video understanding remains focused on offline settings [57, 69, 62, 34], where the model has full access to the complete video sequence before inference. Although this setting facilitates global semantic modeling, it falls short in streaming scenarios, where real-time understanding of continuously evolving scenes is required. Therefore, the development of models specifically designed for online video understanding is of critical importance.

**Streaming Video Understanding.**   In real-world applications, users increasingly expect MLLMs to support online processing and real-time interaction. This demand has prompted growing interest in the task of streaming video understanding. Recently, several works have explored this emerging area [5, 72, 63, 11, 61, 23]. However, most existing streaming video understanding approaches are primarily designed for streaming dense video captioning [5, 60, 32, 47, 12], focusing solely on summarizing semantic content from visual frames. As a result, they struggle to handle essential tasks such as memory recall and real-time perception, which are critical for comprehensive streaming video understanding. Moreover, in pursuit of computational efficiency, many methods apply aggressive compression to video frame sequences [46, 74, 64], making them unsuitable for complex and dynamic tasks that require fine-grained and real-time spatiotemporal understanding, such as autonomous driving. To address these limitations, our goal is to develop a more generalizable and practical approach for online video understanding. It emphasizes fine-grained spatiotemporal features at the moment of query and supports persistent memory storage based on events.

## 3 Methodology

### 3.1 Streaming Video Understanding Architecture: StreamForest

In this section, we detail our proposed StreamForest. Specifically, the core design of StreamForest lies in Fine-grained Spatiotemporal Window and Persistent Memory Forest, which work in tandem to enable the model to retain long-term memories of past events while supporting real-time perception.

#### 3.1.1 Fine-grained Spatiotemporal Window

To meet the real-time spatiotemporal perception requirements of streaming video understanding, we introduce the Fine-grained Spatiotemporal Window (FSTW). We observe that in practical applications, most of the clues requiring fine-grained spatiotemporal reasoning are concentrated near the time of the question. Therefore, we retain only second-level short-term fine-grained spatiotemporal features.

Specifically, the FSTW consists of two components: real-time perception and short-term spatiotemporal memory. Real-time perception directly samples high-resolution visual features from the current frame, which are encoded with spatiotemporal positional information. As new frames arrive, older frames are compressed along the spatial dimension and transferred into short-term spatiotemporal memory. At the same time, the model computes inter-frame similarity between new and old frames to enable subsequent event-level segmentation. The short-term spatiotemporal memory maintains a frame sequence with a duration of $t_s$ seconds. When its capacity is exceeded, overflowing visual features are offloaded into the Persistent Event Memory Forest. We segment continuous visual features into meta-events by identifying the position with the local minimum inter-frame similarity in the frame sequence. This ensures that each meta-event captures a coherent spatiotemporal transition. A meta-event is treated as an independent node, which consists of a collection of visual tokens from similar consecutive frames. These nodes form the foundation of the MLLM's long-term memory.

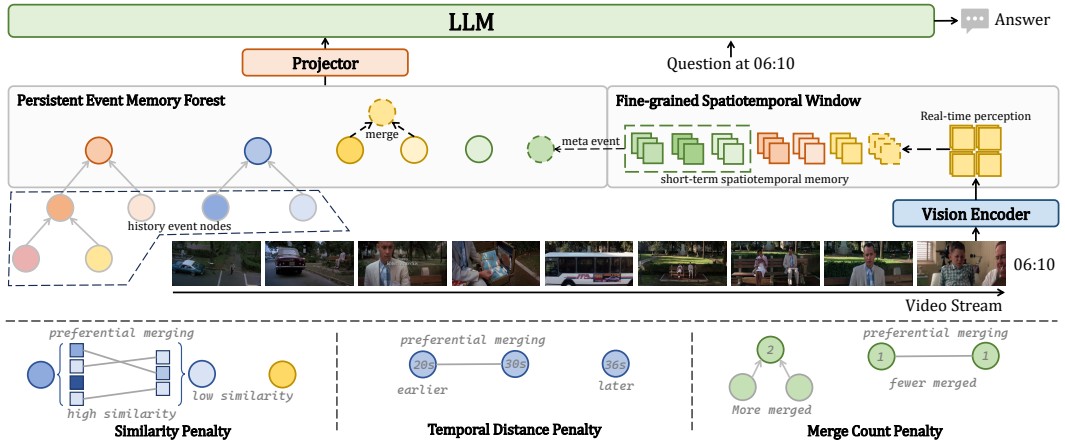

Figure 2: Overview of our proposed StreamForest. The Fine-grained Spatiotemporal Window captures instance-level spatiotemporal features, while the Persistent Event Memory Forest adaptively organizes event-level representations into a set of tree structures. Dashed arrows and feature tokens illustrate potential operations performed during each memory update iteration.

### 3.1.2 Persistent Event Memory Forest

To efficiently process continuously arriving video frame features in streaming scenarios, we propose the Persistent Event Memory Forest (PEMF), a memory architecture specifically designed to support long-term memory in the context of streaming video. Unlike prior methods that rely on direct inter-frame similarity compression [52] or static memory hierarchies [23], PEMF adaptively compresses and organizes video information at the event level. It constructs a hierarchical, tree-structured memory guided by three penalty functions, enabling the model to retain semantically rich and non-redundant content while managing memory efficiently as it evolves over time. To control memory growth, we impose an upper limit $L_q$ on the number of long-term memory tokens stored in PEMF. When this limit is exceeded, PEMF performs hierarchical memory consolidation by adaptively merging adjacent event nodes into single nodes within the tree structure. The selection of nodes for merging is guided by three penalty functions that account for temporal distance, content similarity, and merge frequency, ensuring that the memory remains both informative and compact.

***Similarity Penalty.*** In long videos, adjacent video segments often exhibit high visual similarity, resulting in substantial feature redundancy. Therefore, we introduce a similarity penalty that encourages the merging of event nodes with highly similar visual content. Due to differences in event durations, two candidate event nodes (denoted as $x_i$, $x_{i+1}$) may contain different numbers of visual tokens. To handle this discrepancy, inspired by ToMe [3], we adopt a bipartite graph matching approach. Specifically, we treat the visual features of the two event nodes as sets in a bipartite graph and compute pairwise similarities between tokens across these sets. Let $X_i \in \mathbb{R}^{n_i \times d}$ denote the visual token features of the event node $x_i$, where $n_i$ is the number of tokens in $x_i$. We compute the pairwise cosine similarity matrix $S_i = sim(X_i, X_{i+1}) \in \mathbb{R}^{n_i \times n_{i+1}}$, and select the top $k_i$ highest similarity scores, corresponding to the most similar token pairs between two event nodes. The similarity penalty $P_s$ is defined as one minus the average of these $k_i$ highest similarity scores:

$$P_s(x_i, x_{i+1}) = 1 - \frac{1}{k_i} \sum_{(p,q) \in \mathcal{T}_i} S_i^{(p,q)}, \quad \mathcal{T}_i = \arg\mathrm{TopK}_{(p,q)}(S_i^{(p,q)}, k_i). \tag{1}$$

***Merge Count Penalty.*** When event nodes repeatedly participate in tree-structured hierarchical memory merging, their visual details may gradually degrade due to accumulated information loss. This degradation can lead to local spatiotemporal inconsistencies, ultimately impairing the accuracy of long-term video understanding. To mitigate this issue, we introduce a merge count penalty as a regularization term. It penalizes overly merged nodes and encourages a more balanced memory integration process, thereby preserving the fidelity of each event representation. Let $c_i$ denote the historical merge count of the event node $x_i$, with its maximum value at the query time denoted as

$c_{max}$. We define the merge count penalty $P_m$ as follows:

$$P_m(x_i, x_{i+1}) = \frac{c_i + c_{i+1}}{2c_{max}}. \tag{2}$$

***Temporal Distance Penalty.*** In real-world streaming video understanding scenarios, frames that are temporally closer to the current query time often carry more relevant information. This observation suggests that recent visual features should be preserved with higher fidelity, while historical features can be compressed more aggressively. To implement this intuition, we introduce a temporal distance penalty, which encourages the model to retain more detailed representations of temporally proximate events while promoting the forgetting of details from distant past events. Let $t_q$ denote the current query or interaction time, $t_i$ denote the time of event $i$. The calculation of $t_i$ is detailed in the Appendix A. We define the time penalty $P_t$ as follows:

$$P_t(x_i, x_{i+1}) = 1 - \frac{d_i + d_{i+1}}{2}, \quad d_i = \frac{t_q - t_i}{t_q}. \tag{3}$$

***Overall penalty.*** We incorporate the above three penalties to guide the adaptive merging process of event nodes in the PEMF, where the combination of these three factors determines the merge priority of event node pairs.

$$P(x_i, x_{i+1}) = w_s P_s(x_i, x_{i+1}) + w_m P_m(x_i, x_{i+1}) + w_t P_t(x_i, x_{i+1}). \tag{4}$$

The penalty weights $w_s$, $w_m$, and $w_t$ collectively determine the behavior of PEMF. When only the similarity penalty is applied, the strategy degenerates into similarity-based compression. Using the merge count penalty alone leads to behavior similar to uniform downsampling. When the temporal distance penalty is used in isolation, the method approximates FIFO. By adjusting these penalty weights, our method enables a flexible trade-off among these strategies, allowing it to adapt effectively to various streaming tasks, enabling a balance between efficient storage saving and the retention of task-relevant information across diverse real-world scenarios.

The nodes selected for merging are determined by identifying the pair with the lowest overall penalty score. We employ ToMe [3] for the merging process, compressing the number of visual tokens to half the total tokens of the selected node pair. Upon receiving a user query, the visual features of all root nodes in PEMF, along with all visual features stored in FSTW, are fed into the LLM to support real-time, streaming interaction.

### 3.2 Instruction-tuning Dataset: OnlineIT

Existing offline long video datasets often exhibit distributional bias, where the key evidence for answering questions is typically concentrated in the middle of the video. As a result, MLLMs fine-tuned on such data tend to overemphasize historical content, potentially leading to hallucinations in accurately interpreting the current moment. Although some datasets for streaming video understanding have been released [5, 60, 63, 47], they remain limited in terms of data volume, quality, and task diversity. To address these limitations, we construct OnlineIT, a training dataset specifically designed for streaming video understanding. OnlineIT focuses on fine-grained event comprehension and real-time spatiotemporal understanding in streaming settings, and it significantly enhances the performance of MLLMs on streaming video understanding tasks.

**OnlineIT-general.** Based on criteria of diversity, length, and difficulty, we curated and refined several existing high-quality fine-tuning datasets of streaming video understanding [23, 60, 47]. Building upon these, we further developed two new datasets comprising 32K high-quality streaming training instances. This dataset features a larger scale, broader distribution, and greater task diversity, facilitating the learning of more generalizable streaming video representations.

**OnlineIT-drive.** It includes 89K streaming QA training instances from autonomous driving scenarios. This dataset is designed to enhance MLLMs' performance on complex, real-time downstream tasks. Specifically, by integrating scene semantics, traffic regulations, and common driving events, we extract key elements from driving scenes and video clips to generate a question-answer dataset grounded in autonomous driving contexts. OnlineIT-Drive primarily covers four areas: (1) real-time localization and semantic awareness, (2) understanding of static traffic entities, (3) understanding of dynamic traffic entities, and (4) risk event and accident assessment.

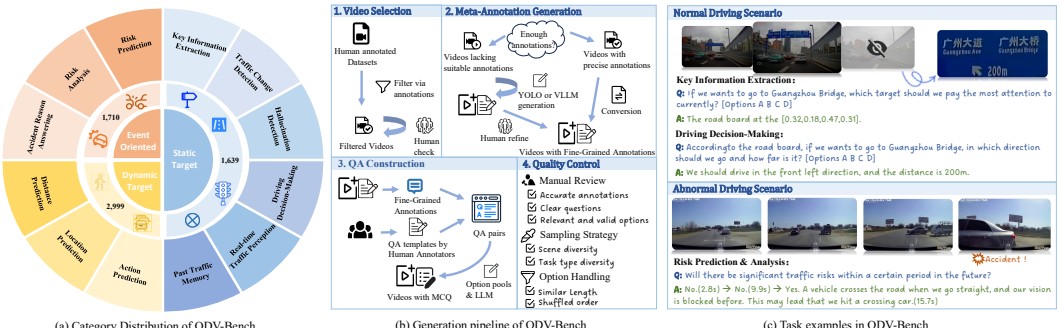

(a) Category Distribution of ODV-Bench     (b) Generation pipeline of ODV-Bench     (c) Task examples in ODV-Bench

Figure 3: (a) The distribution of task types and the number of QA pairs. (b) The detailed pipeline for constructing the ODV-Bench. (c) Typical task examples in ODV-Bench.

# 4 ODV-Bench

Many existing benchmarks for streaming scenarios are derived from offline video evaluation datasets [39, 23, 36, 61], and may not adequately reflect real-world applications of streaming video understanding. Although some of them already incorporate Ego4D videos of daily activities [36, 61], these evaluation samples primarily evaluate MLLMs' ability to perceive static scenes and narrate human-environment interactions in a stepwise manner. In contrast, autonomous driving presents dynamic, high-stakes environments with rapidly changing scenes, complex multi-agent interactions (vehicles, pedestrians, and traffic signals), and demanding prediction tasks (such as risk assessment and motion planning). These scenarios require models to balance long-term event memory with fine-grained short-term perception to avoid accidents and make timely decisions. To address this gap, we introduce ODV-Bench, a benchmark specifically designed for online video understanding in autonomous driving scenarios.

## 4.1 Task Formulation

As shown in Figure 3 (a), we first explore the key traffic elements in autonomous driving scenarios and summarize them into three categories of task scenarios: **(1) Static-target-oriented tasks**, which involve the recognition and retrieval of stationary traffic elements such as traffic signs, lights, and road indicators; **(2) Dynamic-target-oriented tasks**, which focus on behavior and trajectory prediction of dynamic road participants such as vehicles and pedestrians; and **(3) Event-oriented tasks for multi-agent interaction**, which capture complex interactions, risk scenarios, and accidents involving multiple agents. Next, guided by temporal cues and the practical needs of driving, we further define fine-grained task types based on these categories to comprehensively assess model understanding in realistic online driving video scenarios. For more details on task formulation, please refer to Appendix C.1.

## 4.2 Benchmark Construction

The construction process of ODV-Bench is illustrated in Figure 3 (b). We adopt a four-stage approach to ensure the quality of each generated question, and then present some typical task examples across different driving scenarios in Figure 3 (c).

**Data Collection. (1) Video Selection.** To align with real-world driving scenarios, we first curated 6 datasets [24, 14, 82, 66, 73, 4] from different task scenarios within the autonomous driving domain, from normal driving to unexpected events. Then, we designed a semi-automatic pipeline that primarily relies on annotation filtering and YOLO-based detection [25], supplemented by manual inspection, to select task-relevant videos from the collected dataset. **(2) Meta-Annotation Generation.** To obtain meta-annotations with detailed spatiotemporal and semantic information, we developed tailored methods based on existing dataset annotations. For well-annotated datasets, we effectively convert existing labels into task-specific meta-annotations. For others, we design a semi-automatic pipeline that begins with coarse annotations generated by VLLM and YOLO [25], followed by structured human verification to ensure quality.

Table 1: **Evaluation results on ODV-Bench.** Our model significantly outperforms state-of-the-art offline and online video MLLMs under zero-shot testing conditions, and achieves further improvements after fine-tuning on driving-domain data.

| Method | Size | #Frames | Static Target | | | | | | | Dynamic Target | | | | Event Oriented | | | | Overall |
|---|---|---|---|---|---|---|---|---|---|---|---|---|---|---|---|---|---|---|
| | | | RTP | HD | KIE | TCD | DDM | PTM | Avg. | AP | LP | DP | Avg. | RP | RA | ARA | Avg. | |
| *Human* | | | | | | | | | | | | | | | | | | |
| Human Agents | - | - | 96.8 | 97.6 | 98.2 | 95.7 | 95.9 | 94.4 | 95.9 | 83.7 | 87.9 | 90.4 | 88.2 | 91.9 | 94.9 | 93.0 | 92.5 | 91.4 |
| *Open-source Offline Video MLLMs* | | | | | | | | | | | | | | | | | | |
| MiniCPM-V2.6 [65] | 7B | 64 | 20.0 | **87.8** | 15.1 | 49.1 | 26.4 | 20.6 | 27.3 | 71.2 | 73.4 | 47.2 | 60.0 | **73.4** | 33.3 | 16.7 | 53.6 | 49.8 |
| LongVA [75] | 7B | 64 | 29.9 | 7.3 | 37.7 | 47.3 | **38.0** | 33.6 | 31.8 | 66.6 | 58.6 | **50.9** | 56.6 | 57.5 | 58.1 | 46.2 | 56.7 | 50.2 |
| LLaVA-Onevision [28] | 7B | 64 | 36.0 | 4.9 | 22.6 | 60.0 | 31.4 | 39.0 | 34.2 | 53.6 | 70.3 | 47.4 | 55.1 | 57.9 | 72.2 | 47.4 | 62.2 | 51.6 |
| InternVL2.5 [8] | 8B | 32 | 40.1 | 16.3 | 37.7 | 52.7 | 30.4 | 40.9 | 37.2 | 64.1 | **84.6** | 49.5 | **62.5** | 54.0 | 60.6 | 50.6 | 56.1 | 54.2 |
| VideoChat-Flash [33] | 7B | 256 | 29.6 | 15.5 | 45.3 | **76.4** | 26.1 | 36.1 | 32.2 | **73.5** | 75.3 | 47.2 | 61.0 | 67.1 | 64.8 | 46.2 | **64.3** | 54.4 |
| Qwen2.5-VL [2] | 7B | 1fps | **51.8** | 8.1 | **79.3** | 49.1 | 36.0 | **57.3** | **48.3** | 50.4 | 82.6 | 46.9 | 57.5 | 47.6 | **78.6** | **52.6** | 59.4 | **55.6** |
| *Open-source Online Video MLLMs* | | | | | | | | | | | | | | | | | | |
| Flash-VStream [72] | 7B | 1fps | 25.4 | 1.6 | 11.3 | 50.9 | 36.0 | 22.1 | 24.8 | 25.5 | 39.8 | 47.2 | 40.2 | 32.4 | 48.6 | 30.1 | 38.1 | 35.7 |
| Dispider [47] | 7B | 1fps | 31.1 | 7.3 | 34.0 | 63.6 | 34.0 | 35.4 | 32.5 | 43.2 | 73.1 | 45.8 | 52.7 | 38.2 | 55.4 | 36.5 | 44.3 | 45.2 |
| VideoChat-Online [23] | 4B | 1fps | 36.9 | 0.8 | 62.3 | 49.1 | 21.5 | 47.0 | 36.1 | 70.2 | 86.7 | 46.4 | 62.9 | 51.2 | 69.4 | 45.5 | 57.4 | 54.5 |
| **StreamForest** | **7B** | **1fps** | 51.4 | 15.5 | 54.7 | 56.4 | **38.6** | 65.3 | 51.5 | 72.6 | 83.2 | 46.0 | 62.3 | 60.2 | 73.3 | 47.4 | 63.8 | 59.9 |
| **StreamForest (FT-drive)** | **7B** | **1fps** | **70.1** | 17.1 | **100.0** | 60.0 | 32.7 | **83.6** | **64.6** | 64.0 | **96.6** | **59.6** | **70.7** | **71.8** | **93.4** | **58.3** | **78.5** | **71.2** |

**QA Construction. (1) MCQ Generation.** To enable efficient automatic QA generation, we first design accurate and diverse templates tailored to each defined task. These templates are then populated with fine-grained and precise annotations to generate high-quality QA pairs. Next, we develop a multiple-choice generation pipeline based on an option pool, introducing plausible yet misleading distractors alongside the correct answer to ensure the realism and effectiveness of choices. **(2) Quality Control.** To ensure benchmark quality, we first conduct multiple rounds of manual review to verify the clarity and accuracy of QA pairs and the plausibility of distractor options. Besides, to enhance scene diversity and task-type balance, we apply a sampling strategy that allocates questions proportionally to video length, maximizing coverage across scenarios.

# 5 Experiments

**Implementation Details.** We adopt SigLiP-so400M[70] as the visual encoder, use an MLP as the projection head, and we employ Qwen2-7B as the LLM. By default, the number of visual tokens is capped at 8192. Among these, 729 tokens are allocated to real-time perception, while short-term spatiotemporal memory consists of 18 frames, each represented by 128 visual tokens. We set the penalty weights for similarity, merge count, and temporal distance to 0.4, 0.4, and 0.2, respectively. The model is trained on 32 A100 GPUs using our proposed OnlineIT dataset, supplemented with offline video data from VideoChat-Flash [33] and LLaVA-Video [76], as well as image data from LLaVA-OneVision [28]. We adopt a five-stage training strategy to train StreamForest from scratch. The first three stages follow the training paradigm of offline long video MLLMs [33]. The fourth stage performs streaming video fine-tuning to yield the base StreamForest. In addition, an optional fifth stage can be incorporated by training with the OnlineIT-Drive, which yields the StreamForest(FT-drive). During the evaluation phase, we constrain the model to process streaming frames at 1 FPS. For more detailed implementation specifics, please refer to the Appendix D.

## 5.1 Online Benchmark Results

We evaluate the performance of our model on four benchmarks for online video question answering: ODV-Bench, StreamingBench [39], OVBench [23], and OVO-Bench [36]. These benchmarks follow streaming video QA scenarios, where the VideoLLMs must process only the video content available before the current timestamp.

**ODV-Bench.** It closely integrates spatiotemporal information to comprehensively evaluate MLLMs' ability to understand fine-grained details in online videos and to make future predictions based on both historical and current context in autonomous driving scenarios. The benchmark includes tasks such as identifying subtle objects or actions, describing object positions and spatial relations, and

Table 2: **Comparison of our method with existing approaches on video question answering tasks across various scenarios.** Our approach significantly outperforms previous methods on streaming video understanding benchmarks, while maintaining strong and competitive performance on both long and short video understanding.

| Method | Size | Online Video | | | Long Video | | Short Video | |
| | | StreamingBench | OVBench | OVO-Bench | VideoMME | MLVU | MVBench | PerceptionTest |
| | | Real-Time All | Avg | Overall | w/o sub. | M-Avg | Avg | Val |
|---|---|---|---|---|---|---|---|---|
| *Open-source Offline Video MLLMs* | | | | | | | | |
| InternVL2 [9] | 8B | 63.7 | 48.7 | 50.1 | 54.0 | 64.0 | 65.8 | - |
| LongVA [75] | 7B | 60.0 | 43.6 | - | 52.6 | 56.3 | - | - |
| LLaVA-OneVision [28] | 7B | 71.1 | 49.5 | 52.9 | 58.2 | 64.7 | 56.7 | 57.1 |
| Qwen2-VL [55] | 7B | 69.0 | 49.7 | 52.7 | 63.3 | - | 67.0 | 66.9 |
| LongVU [50] | 7B | - | - | 48.5 | 60.6 | 65.4 | 66.9 | - |
| LLaVA-Video [76] | 7B | - | - | 53.1 | 63.3 | 70.8 | 58.6 | 67.9 |
| *Open-source Online Video MLLMs* | | | | | | | | |
| VideoLLM-online [5] | 8B | 36.0 | 9.6 | 12.8 | - | - | - | - |
| MovieChat [52] | 7B | - | 30.9 | - | 38.2 | - | 55.1 | - |
| Flash-VStream [72] | 7B | 23.2 | 31.2 | 33.2 | - | - | - | - |
| VideoChat-Online [23] | 4B | - | 54.9 | - | 52.8 | - | 64.9 | - |
| Dispider [47] | 7B | 67.6 | - | 41.8 | 57.2 | 61.7 | - | - |
| **StreamForest** | **7B** | **77.3** | **60.5** | **55.6** | **61.4** | **70.0** | **70.2** | **73.1** |
| **StreamForest (FT-drive)** | **7B** | **76.8** | **61.6** | **55.6** | **61.9** | **69.6** | **68.6** | **71.6** |

forecasting object trajectories. These tasks require strong real-time spatiotemporal perception and contextual understanding. As shown in Table 1, StreamForest achieves an average accuracy of 59.9% on ODV-Bench without being trained on OnlineIT-drive and further improves to 71.2% after training on it. This significantly outperforms all existing online and offline MLLMs, demonstrating the strong generalization capability of our method to downstream streaming video understanding tasks. These results highlight its potential for real-world applications.

**StreamingBench & OVBench & OVO-Bench.** As shown in Table 2, StreamForest demonstrates strong performance across existing open-source streaming video understanding benchmarks. It achieves an accuracy of 77.3% on StreamingBench, 60.5% on OVBench, and 55.6% on OVO-Bench. These impressive results highlight the robustness of StreamForest in a wide range of online video understanding scenarios. The superior performance of our model can be attributed to two key architectural innovations. First, the Fine-grained Spatiotemporal Window enables precise spatial perception and responsive short-term temporal modeling, which are critical for real-time perception and forward responding tasks. Second, the Persistent Event Memory Forest adaptively organizes long-term visual content into a structured and efficient memory forest, significantly enhancing the MLLM's ability to retain and reason over past events. Together, these two modules offer complementary capabilities that allow our model to handle dynamic, long-horizon streaming video inputs effectively, while maintaining high contextual coherence.

## 5.2 Offline Benchmark Results

We further evaluate our method on two long video understanding benchmarks (VideoMME[16] and MLVU[77]) and two short video datasets (MVBench[31] and PerceptionTest[45]). In the offline setting, the entire video is provided as input to the MLLM. We sample video frames at 1 FPS, with a maximum limit of 2048 frames. For videos exceeding this limit, frames are uniformly sampled across the entire duration. As shown in Table 2, our method demonstrates superior performance on both long and short video understanding tasks compared to recent state-of-the-art online Video MLLMs. In addition, it outperforms leading offline models in most benchmarks, achieving 61.4% on VideoMME, 70.0% on MLVU, 70.2% on MVBench, and 73.1% on PerceptionTest. This strong performance in offline scenarios highlights the robust generalization capability of our proposed method.

## 5.3 Ablations

**Effectiveness of the Persistent Event Memory Forest:** We replace the proposed PEMF with several methods used in previous work. To ensure a fair comparison, we keep the visual token budgets consistent across all methods and fine-tune each model accordingly. The ablation results

Table 3: Comparison between our proposed PEMF and other commonly used memory strategies.

| Memory Policy | OVBench | OVO-Bench | MLVU |
|---|---|---|---|
| | Avg | Overall | M-Avg |
| Uniform Sampling | 58.2 | 52.7 | 69.4 |
| First In First Out | 58.7 | 52.9 | 56.7 |
| Similarity Merge [52] | 60.3 | 53.4 | 68.0 |
| Pyramid Memory Bank [23] | 60.3 | 53.9 | 68.2 |
| **PEMF (Ours)** | **60.5** | **55.6** | **70.0** |

Table 4: Ablation study on the key components of StreamForest.

| Model | OVBench | OVO-Bench | MLVU |
|---|---|---|---|
| | Avg | Overall | M-Avg |
| w/o FSTW & PEMF | 58.0 | 52.5 | 51.8 |
| w/o FSTW | 59.1 | 53.7 | 69.4 |
| w/o PEMF | 58.9 | 53.5 | 56.6 |
| w/o Event | 59.4 | 52.6 | 69.1 |
| **Ours** | **60.5** | **55.6** | **70.0** |

are shown in Table 3. The FIFO strategy shows the worst performance. This is especially evident on the long-video benchmark MLVU (56.7% vs. 70.0%), where the method fails due to unfiltered discarding of historical visual features. OVBench primarily emphasizes short-term, fine-grained spatiotemporal perception. Uniform sampling reduces the resolution of recent visual information, which is crucial for real-time understanding (58.2% vs. 60.5% on OVBench). Similarity Merge achieves performance comparable to our PEMF on OVBench (60.3% vs. 60.5%). However, its limitations become clear in tasks that require persistent memory and long-horizon reasoning. On OVO-Bench, PEMF outperforms Similarity Merge by +2.2%, and on MLVU by +2.0%. This is because similarity-based merging may over-merge frames within local video segments, potentially leading to spatiotemporal irregularities and the loss of local event-level representations. The pyramidal memory bank maintains memory through frame replacement. However, fixed capacity limits its ability to capture long-range spatiotemporal features (53.9% vs. 55.6% on OVO-Bench and 68.2% vs. 70.0 on MLVU). In contrast, our method evaluates each visual event based on event-level similarity, merge count, and temporal distance. Then it performs memory consolidation at the event level. This strategy supports efficient and persistent maintenance of historical visual features.

**Effectiveness of the Overall Architecture:** We conduct ablation studies on three key architectural components. Specifically, we ablate the Fine-grained Spatiotemporal Window and the Persistent Event Memory Forest, while ensuring that the total number of visual tokens remains consistent with the original configuration. In addition, we replace event-based node construction with a frame-based approach. As shown in Table 4, removing both modules leads to the most significant performance drop. Using either FSTW or PEMF alone improves performance compared to the baseline, but the best results are achieved when both components are integrated (+2.5% on OVBench, +3.1% on OVO-Bench, and +18.2% on MLVU). This joint ablation confirms that FSTW and PEMF provide complementary benefits. FSTW enhances real-time spatiotemporal perception near the query timestamp, while PEMF supports efficient and persistent long-term memory, together yielding the strongest overall performance. Moreover, event-level node construction effectively prevents over-merging within events, enabling the compression of visual features at the level of complete visual events rather than individual frames.

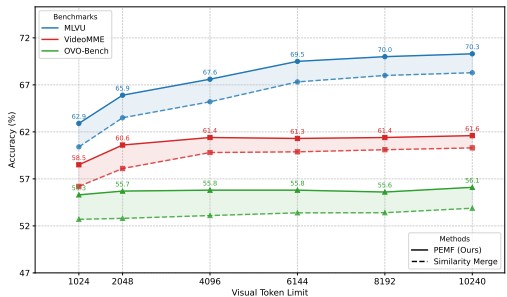

Figure 4: Performance under varying visual token budgets.

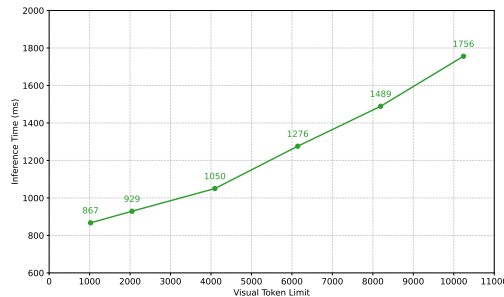

Figure 5: Average inference time under varying visual token budgets (on single A100 GPU).

**Robustness to Different Visual Token Budgets:** We evaluate the robustness of our method under varying budget constraints for visual tokens, ranging from 1K to 10K. Figure 4 illustrates the performance variation on the benchmarks MLVU, VideoMME, and OVO-Bench under these settings. Notably, under the strict constraint of only 1K visual tokens, StreamForest achieves an average

Table 5: Quantitative analysis of runtime and memory usage of StreamForest.

| Input Frames | Memory (GB) | FLOPs (T) | Latency (s) |
|---|---|---|---|
| 64 | 15.8 | 93.1 | 0.776 |
| 256 | 17.1 | 134.1 | 1.126 |
| 1024 | 17.2 | 137.3 | 1.497 |

Table 6: Runtime comparison of PEMF with other memory mechanisms for 500 frames.

| Method | Vis. Encode (s) | Mem. Update (s) | LLM infer (s) | Total (s) |
|---|---|---|---|---|
| Similarity Merge [52] | 5.198 | 0.183 | 1.388 | 6.769 |
| PMB [23] | 5.203 | 0.451 | 1.381 | 7.035 |
| **PEMF (Ours)** | 5.218 | **0.172** | 1.394 | 6.784 |

visual compression ratio of up to 99.8% on long video benchmarks. Despite this extreme level of compression, the model still maintains competitive performance, which strongly demonstrates the robustness of our approach in persistently preserving long-term, event-level visual memories. We also conduct a direct comparison between our PEMF and the Similarity Merge. The results clearly demonstrate two core benefits of our approach. First, PEMF exhibits superior absolute performance, consistently outperforming Similarity Merge across all token budgets with an average accuracy improvement of 2-3%. Second, our method shows stronger resilience to extreme compression. Under the most severe 1K token budget, PEMF retains a higher fraction of its full-budget performance, achieving a notable +1.8% relative retention advantage on VideoMME. These experimental results confirm that the performance gains stem from the intrinsic design of PEMF, which adaptively consolidates event-level memory to preserve semantically salient information, thereby ensuring both high accuracy and token efficiency under stringent resource constraints. Figure 5 presents the average inference time of StreamForest under different budgets of visual tokens. The fast and stable inference speed highlights the practical applicability of StreamForest.

**Computational cost:** We provide a quantitative analysis of the runtime and memory usage of StreamForest. To isolate the effect of the memory mechanism, we assume that frame-level visual features are already extracted by the vision encoder in real time. As shown in Table 5, PEMF enforces a strict upper bound of visual tokens (8K here), ensuring that memory usage remains stable ($\bar{1}$7 GB) regardless of the number of processed frames. Consequently, the FLOPs and inference latency do not grow significantly even with longer streaming inputs. In addition, we compare PEMF with other memory mechanisms, including the Pyramid Memory Bank [23] and the Similarity Merge strategy [52]. As summarized in Table 6, the overall runtime is dominated by vision encoding (5.2s for 500 frames, ~95 FPS). The memory update of PEMF is extremely lightweight (0.172s for 500 frames), which is negligible compared to the efficiency gains achieved by significantly reducing the total number of visual tokens.

**Impact of Training Data:** Table 7 presents the impact of our training strategy that integrates both online and offline datasets. The results clearly demonstrate that combining OnlineIT with existing offline VideoQA datasets significantly improves performance on streaming video understanding benchmarks. OnlineIT is specifically designed for real-time perception and future prediction in streaming scenarios, effectively mitigating hallucinations caused by inconsistencies between historical spatiotemporal context and the current moment.

Table 7: The contribution of our training data to performance on the streaming video understanding benchmark. O.general refers to OnlineIT-general, while O.drive refers to OnlineIT-drive.

| Data | ODV-Bench | OVBench | OVO-Bench |
|---|---|---|---|
| | Avg | Avg | Overall |
| base | 56.3 | 53.9 | 53.5 |
| base + O.general | 59.9 | 60.5 | **55.6** |
| base + O.general + O.drive | **71.2** | **61.6** | **55.6** |

## 6 Conclusions

In this work, we have proposed StreamForest, a novel architecture for streaming video understanding that addresses the limitations in long-term memory and fine-grained perception. By introducing the Persistent Event Memory Forest, our method effectively manages historical visual information through adaptive merging guided by temporal distance, content similarity, and merge count penalties. Coupled with the Fine-grained Spatiotemporal Window, the model maintains a precise understanding of the current scene. We also present OnlineIT, a streaming video understanding fine-tuning dataset that mitigates spatiotemporal shift issues and enhances real-time perception and reasoning. As well as ODV-Bench, a new benchmark tailored for real-time autonomous driving scenarios. Extensive experiments demonstrate that StreamForest not only outperforms state-of-the-art streaming video MLLMs but also rivals top offline video MLLMs under strict streaming input settings, showcasing its robustness and practical value in real-time streaming video understanding applications.

## Acknowledgement

This work is supported by the National Key R&D Program of China (No. 2022ZD0160900), the Basic Research Program of Jiangsu (No. BK20250009), and the Collaborative Innovation Center of Novel Software Technology and Industrialization.

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

# A  Details of node's timestamp

Each event node's timestamp is initialized as the average time of the frames it represents (e.g., if an event node spans frames from 10s to 14s, its timestamp is initialized as 12s). When two event nodes are merged, the timestamp of the new node is computed as a token-count-weighted average of the original nodes:

$$t_{\text{new}} = \frac{t_i \cdot n_i + t_j \cdot n_j}{n_i + n_j}, \tag{5}$$

where $t_i$, $t_j$ are the timestamps of the original event nodes, and $n_i$, $n_j$ are the numbers of visual tokens contained in each node, respectively. This weighted scheme prevents timestamp drift during multiple rounds of merging, especially when the merged nodes contain significantly different amounts of visual tokens.

# B  Details of OnlineIT

In this section, we provide a comprehensive description of the task categorization and data distribution of the OnlineIT dataset. It is specifically designed to enhance the streaming video understanding capabilities of MLLMs in terms of real-time perception, future prediction, and event continuity. As shown in Table 8, the dataset is divided into two major components: OnlineIT-general, which targets general streaming video understanding, and OnlineIT-drive, which focuses on autonomous driving scenarios. Each subset is carefully designed to cover a diverse range of fine-grained perception and reasoning tasks with high-quality annotations.

Table 8: Task types and data volumes of OnlineIT.

| Dataset | Categories | Task | Source | Instance Num |
|---|---|---|---|---|
| OnlineIT-general | Spatial Perception | Spatial Grounding | RefCOCO [67] | ~43k |
| | | | Allseeing-V2 [56] | ~45k |
| | | Multi-round Spatial Understanding | Visual Genome [27] | ~43k |
| | | Spatial Grounded VQA | Allseeing-V2 [56] | ~43k |
| | | Relative Spatial Localization | LaSOT [13] | ~19k |
| | Temporal Perception | Temporal Grounding | Charades-STA [17] | ~11k |
| | | | HiREST [68] | ~0.4k |
| | | | QuerYD [43] | ~13k |
| | | Reasoning Temporal Localization | ActivityNet-RTL [20] | ~10k |
| | | Multi-format Temporal Grounding | InternVid-VTime [19] | ~20k |
| | Spatiotemporal Perception | Spatiotemporal Action Localization | AVA [18] | ~6k |
| | | Object Backward Tracking | LaSOT [13] | ~51k |
| | | | GOT [22] | ~58k |
| | | Spatiotemporal Detection | LaSOT [13] | ~14k |
| | Event Perception | Dense Video Captioning | ActivityNet-Captions [26] | ~10k |
| | | | ViTT [78] | ~5k |
| | | | Youcook2 [21] | ~1k |
| | | Step Localization and Captioning | COIN [53] | ~9k |
| | | | HiREST [68] | ~0.5k |
| OnlineIT-drive | Static Target | Past Memory | D²-City [4] | ~9k |
| | | Real-time Perception | TT100k [80] | ~46k |
| | | | D²-City [4] | ~13k |
| | Dynamic Target | Localization Prediction | Road-waymo [24] | ~1k |
| | | Move Distance Prediction | Road-waymo [24] | ~7k |
| | Event Oriented | Accident Reasoning | MM-AU [14] | ~6k |
| | | Risk Analysis | MM-AU [14] | ~7k |

## B.1  OnlineIT-general

OnlineIT-general encompasses a broad scope of tasks designed to foster a comprehensive understanding of spatiotemporal video content in streaming settings. As shown in Table 8, the dataset is categorized into four primary task types: spatial perception, temporal perception, spatiotemporal perception, and event perception. To ensure diversity, robustness, and fine-grained task coverage, we

compiled and refined data from a wide array of sources. In total, OnlineIT-general comprises over 400k instances spanning various difficulty levels and video durations.

**Spatial Perception.** This task type includes four subtasks. ***Spatial Grounding*** requires the model to output the bounding box indicating the location of a queried object. ***Multi-round Spatial Understanding*** involves identifying the object's spatial location through multi-turn dialogue or generating a caption for the object within a specified spatial region. ***Spatial Grounded VQA*** combines visual question answering with spatial localization, requiring the model to provide the bounding box of the relevant area while answering the question. ***Relative Spatial Localization*** challenges the model to determine the position of a specified object relative to the overall scene. These tasks emphasize spatial grounding and reasoning, which are crucial for enhancing a model's fine-grained spatial perception in real-time streaming video scenarios.

**Temporal Perception.** This category consists of three subtasks. ***Temporal Grounding*** involves interpreting a natural language query and identifying the start and end timestamps of the corresponding video segment. In streaming scenarios, the model must also assess whether the described event is currently ongoing. ***Reasoning Temporal Localization*** requires identifying the relevant time span of an event while answering a reasoning-based question. ***Multi-format Temporal Localization*** incorporates both single-turn and multi-turn dialogues, covering a diverse range of question formats. These tasks focus on strengthening the MLLM's ability to track and reason about temporal dependencies, improving its understanding of both current and past moments in a video stream.

**Spatiotemporal Perception.** This task type integrates spatial and temporal reasoning and includes three subtasks. ***Spatiotemporal Action Localization*** requires the model to predict both the spatial location and the action being performed by a target at a specific query time. ***Object Backward Tracking*** tasks the model with identifying the current location of an object and tracing its position at previous time points, such as one or two seconds earlier. ***Spatiotemporal Detection*** operates over broader temporal windows, asking whether an object visible in the current frame existed several seconds ago or requiring the model to locate an object at a specified historical moment and determine its duration of existence. These tasks combine spatiotemporal cues to capture actions, motion, and transitions, allowing the model to track object trajectories and anticipate future states based on past and present context.

**Event Perception.** This category includes two subtasks. ***Dense Video Captioning*** involves detecting a sequence of events in a video and generating corresponding timestamps along with high-level descriptions. ***Step Localization and Captioning*** differs by focusing on segmenting and narrating key procedural steps within long-form videos. These tasks are aimed at improving the model's understanding of complex, multi-step events, enabling structured interpretation of dynamic sequences in streaming video understanding.

### B.2 OnlineIT-drive

OnlineIT-drive is designed specifically for the domain of streaming video understanding in autonomous driving. The dataset includes 89k instances, which are organized into three major task categories. Collectively, these tasks aim to strengthen not only real-time perception capabilities but also the temporal reasoning and decision-making abilities of MLLMs in high-stakes and rapidly evolving environments.

**Static Target Understanding.** To improve the model's capacity for static scene understanding, two task types are introduced. ***Real-time Perception*** requires the model to accurately perceive and interpret the semantics and spatial attributes of traffic-related targets as they appear in real time. ***Past Memory*** assesses the model's ability to retain and retrieve the semantics and spatiotemporal characteristics of traffic targets that were observed at a prior point in time. These tasks collectively enhance the model's capability to perceive, understand, and remember static traffic elements and environmental context, such as road infrastructure and regulatory signage.

**Dynamic Target Understanding.** It includes two task types that aim to enhance predictive understanding of dynamic traffic participants. ***Location Prediction*** requires the model to estimate

the future position of a moving target based on its historical motion trajectory. ***Move Distance Prediction*** focuses on predicting the distance traveled between the ego vehicle and other moving agents, given motion-related observations. These tasks are designed to improve the model's ability to track continuously moving objects and to anticipate future trajectories.

**Event Oriented Reasoning.** It is intended to foster the development of reasoning abilities necessary for risk assessment and accident interpretation. ***Risk Analysis*** requires the model to detect potential sources of danger in the current traffic scene and to assess the likelihood of accident occurrence. ***Accident Reasoning*** involves post hoc analysis, where the model must infer the causes of an observed accident and articulate plausible preventive strategies. These tasks are designed to enhance the model's ability to reason about causal relationships and to anticipate or reflect on traffic risks with contextual awareness.

# C   Details of ODV-Bench

In this section, we detail the task taxonomy and formulation of the ODV-Bench, as well as the dataset statistics. We categorize task scenarios based on target entities and derive key perception and reasoning task types for each scenario in Table 9.

Table 9: Overview of task categories, their subcategories, and question templates.

| Task Objective Scenario | Sub-task | Query Examples |
|---|---|---|
| Static Target | Real-time Traffic Perception | 1) What is the meaning of the traffic sign at the [0.61,0.31,0.64,0.38] in the current picture? 
 2) What are the position coordinates of the traffic sign indicating "Pedestrian Crossing" in the current picture? 
 3) What is the meaning of the road board at the [0.77,0.08,0.88,0.2] in the current picture? 
 4) According to the road board at the [0.09,0.46,0.19,0.52] currently, if going in the left direction, where will we go, how far is it? 
 5) What is the color of the traffic light at the [0.42,0.01,0.45,0.13] in the current picture? And what is its indication? 
 6) According to the road board at the [0.65,0.03,0.76,0.16] currently, how far is it from Fengle? |
| | Past Traffic Memory | 1) What were the position coordinates of the traffic sign indicating "No Left Turn" in the scene 3 seconds ago? 
 2) What was the meaning of the traffic sign at the [0.39,0.13,0.41,0.17] in the scene 1 seconds ago? 
 3) The traffic sign is currently located at [0.92,0.02,0.96,0.09]. What were its coordinates 2 seconds ago? 
 4) What was the color of the traffic light at the [0.35,0.1,0.38,0.22] in the scene 2 seconds ago? |
| | Driving Decision-Making | 1) According to the road board at the [0.37,0.18,0.47,0.31] in the image taken 2 seconds ago, if we wants to go to Renhe, in which direction should we go and how far is it? 
 2) According to the road board at the [0.51,0.18,0.61,0.3] currently, if we wants to go to Libai Avenue, in which direction should we go and how far is it? 
 3) According to the road board at the [0.38,0.04,0.63,0.23] currently, if we wants to turn left, which lane should we be in? |
| | Key Information Extraction | 1) If we wants to go to Suzhou, which target should we pay the most attention to currently? Provide the type and coordinates. |
| | Hallucination Detection | 1) According to the road board at the [0.38,0.31,0.52,0.48], how far is it currently from Qingpu Town? 
 2) What is the meaning of the traffic sign at the [0.34,0.29,0.38,0.38] in the current picture? 
 3) What is the color of the traffic light at the [0.9,0.86,0.95,0.92] in the current picture? |
| | Traffic Change Detection | 1) At the current moment, has the traffic signal light indicating "turn right" ahead turned completely green? 
 2) At the current moment, has the traffic signal light ahead turned completely red? |
| Dynamic Target | Action Prediction | 1) What will be the subsequent motion state of the car currently in the [0.993, 0.615, 1.0, 0.63] location? |
| | Location Prediction | 1) What will the position box of the pedestrian in the [0.544, 0.561, 0.613, 0.895] location be like in the next second? |
| | Distance Prediction | 1) Is the distance between our car and the car in the [0.488, 0.488, 0.501, 0.494] getting farther or closer? |
| Multi-agent Interaction Event | Risk Prediction | 1) Is there a high probability of traffic accidents occurring within a certain period in the future? 
 2) Will there be significant traffic risks within a certain period in the future? |
| | Risk Analysis | 1) There is a high risk of traffic accidents at present. Based on the environment, what types of accidents are likely to occur, and what is the basis for this prediction? 
 2) There are significant traffic risks at present. Based on the environment, what are the sources of these risks and what types of accidents might they cause? |
| | Accident Reason Answering | 1) What is the cause of the accident in the video? What measures can be taken to avoid it? |

## C.1   Task Taxonomy and Formulation

We first identify the primary categories of traffic entities relevant to autonomous driving and organize task scenarios into three groups: **(1) Tasks for Static Targets**, which involve the recognition and retrieval of stationary traffic elements such as traffic signs, lights, and road indicators; **(2) Tasks for dynamic targets**, which focus on behavior prediction and localization of moving entities such as vehicles and pedestrians; and **(3) Tasks for multitarget interaction events**, which capture complex interactions, risk scenarios, and accidents involving multiple agents. Based on these categories and guided by temporal cues and the practical needs of driving, we further define fine-grained task types to comprehensively assess model understanding in realistic online driving video scenarios.

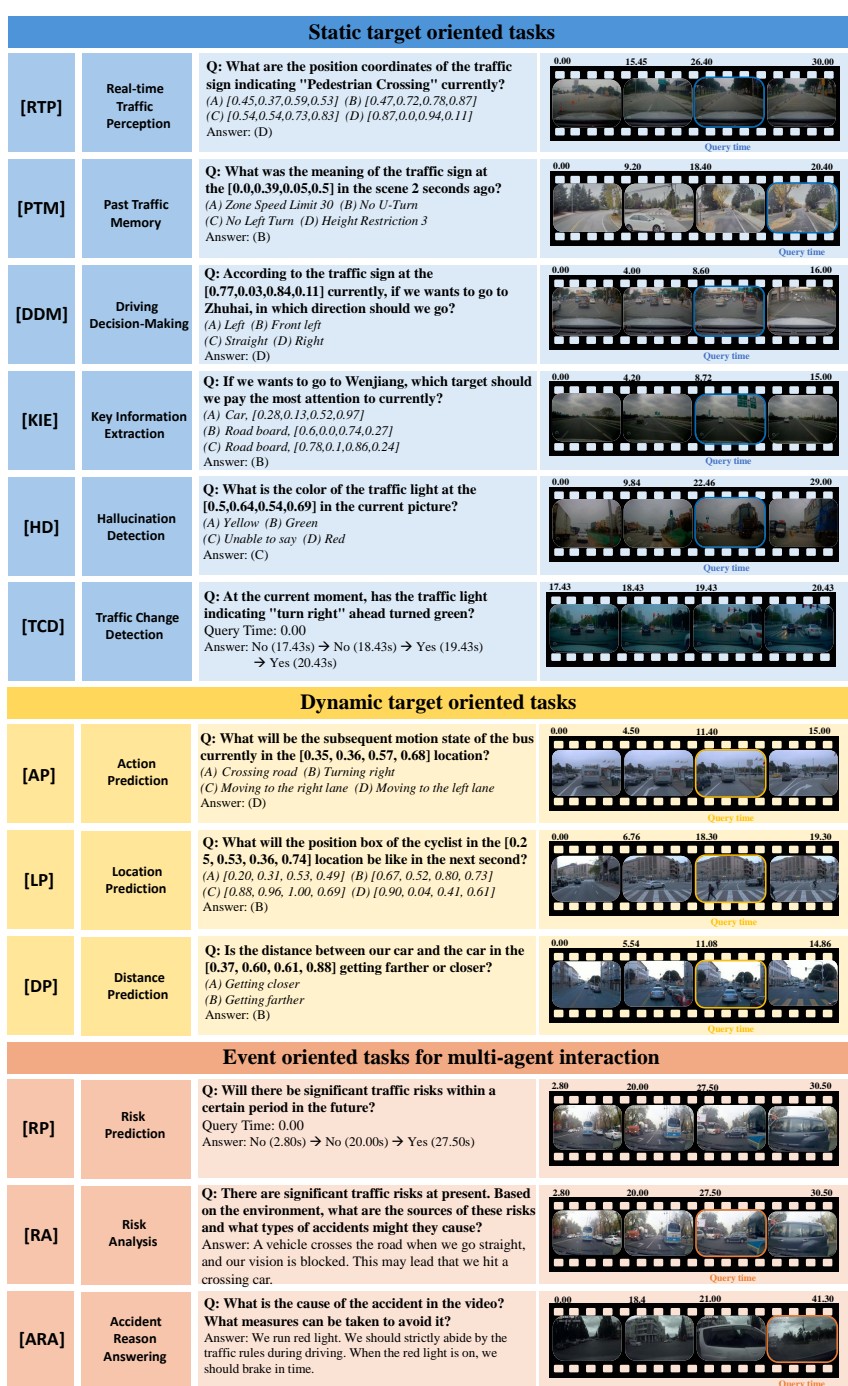

Figure 6: Examples of each task in ODV-Bench. The 12 tasks are divided into three different perception modes for online video understanding for autonomous driving.

### C.1.1 Tasks for Static Targets

Static traffic elements, such as traffic signs and road indicators, play a crucial role in driving decisions and hazard avoidance under normal driving conditions. To evaluate the model's ability to retrieve and recognize these elements in online video streams, we design a dedicated set of tasks. Specifically, we distinguish between basic perception tasks and more advanced reasoning tasks, and further refine them based on temporal cues and practical driving needs: **(1) Real-time Traffic Perception:** Perceive

and interpret the semantics and spatial locations of static traffic elements in real time; **(2) Past Traffic Memory:** Recall and track the semantics and spatiotemporal states of previously observed static elements; **(3) Driving Decision-Making:** make driving decisions based on the perceived information; **(4) Key Information Extraction:** Identify and locate key traffic elements critical to driving decisions; **(5) Hallucination Detection:** identify questions irrelevant to the existing video input; and **(6) Traffic Change Detection:** detect timestamps for changes in traffic elements, such as traffic lights.

### C.1.2 Tasks for Dynamic Targets

The position and behavior of other road participants, such as vehicles and pedestrians, are crucial reference factors influencing autonomous driving decisions and safety. The ability to predict the position and behavior of dynamic traffic objects is essential to ensure the safety of autonomous driving. Therefore, we focus on the following three tasks to effectively evaluate this capability: **(1) Action Prediction:** predicting the next action of vehicles and pedestrians based on continuous spatiotemporal cues; **(2) Distance Prediction:** predicting the relative distance change between the ego-vehicle and other vehicles based on motion information; and **(3) Location Prediction:** predicting the future spatial position of dynamic traffic targets based on their movement trajectories.

### C.1.3 Tasks for Multi-Target Interaction Events

To achieve safe and reliable autonomous driving, the system must be able to identify risks and analyze accidents in complex road interaction scenarios. In the context of online video streams, this ability involves the dynamic recognition and analysis of multi-agent interactions, as well as the reasonable prediction of traffic risks. To evaluate this capability, we design the following three task categories: **(1) Risk Prediction:** predicting the occurrence of significant traffic risks and responding proactively; **(2) Risk Analysis:** detecting the sources of current traffic risks and analyzing the potential causes of accidents; and **(3) Accident Reason Answering:** post-accident analysis, providing potential causes for the incident and summarizing actionable lessons learned.

### C.2 Dataset Statistics

ODV-Bench comprises 1,190 unique first-person driving video clips, encompassing a diverse range of driving scenarios across different countries from routine driving conditions to potential hazards and accidents. The length of videos ranges from 5 seconds to 90 seconds, effectively capturing the diversity of real-world streaming driving experiences. The benchmark includes 6,348 question-answer pairs, with an average query timestamp of 18.9 seconds. Specifically, the static-object-oriented category comprises 247 videos with a total of 1,639 questions; the dynamic-object-oriented category includes 162 videos and 2,999 questions; and the event-oriented category consists of 781 videos with 1,710 questions. All questions are in multiple-choice format, with the number of options varying between 2 and 4 depending on the question type.

## D  More Implementation Details

Table 10: Parameter settings for three-stage offline pre-training.

| | | Stage-1 | Stage-2 | Stage-3 |
|---|---|---|---|---|
| *Vision* | **Resolution×Num. frames** | 384 | 384×8 | Max 384×512 |
| | #Tokens | 64×4 | 64×8 | Max 16×512 |
| *Data* | **Dataset** | Image & Short Video | Image & Short Video | Image & Short / Long Video |
| | #Samples | 0.6M & 0.5M | 3.8M & 3.4M | 0.5M & 2.8M |
| *Model* | **Trainable** | Projector | Full Model | Full Model |
| | #parameters | 16.98MB | 8030.35MB | 8030.35MB |
| *Training* | **Batch Size** | 512 | 256 | 256 |
| | **LR** of *vision encoder* | $1\times10^{-3}$ | $2\times10^{-6}$ | $2\times10^{-6}$ |
| | **LR** *of connector & LLM* | $1\times10^{-3}$ | $1\times10^{-5}$ | $1\times10^{-5}$ |
| | **Epoch** | 1 | 1 | 1 |

Table 11: Parameter settings for the fourth stage online fine-tuning and fifth dirve fine-tuning.

| | | Stage 4 | Stage5 |
|---|---|---|---|
| Data | Dataset | Image & (Short/Long/Online)-Video | Image & (Short/Long/Online)-Video |
| | #Samples | 0.4M & 1.3M | 0.2M & 0.5M |
| Model | Trainable | Projector & LLM | Projector & LLM |
| | #parameters | 7632.60MB | 7632.60MB |
| Vision | Resolution | 384×384 | 384×384 |
| | Frames | 2∼512 | 2∼512 |
| | FPS | 1 | 1 |
| Memory | Real-time Perception Qouta | 729 | 729 |
| | Spatiotemporal Memory Quota | 128 × 18 | 128 × 18 |
| | Total Visual Token Limits | 8192 | 8192 |
| | Similarity Penalty Weight | 0.4 | 0.4 |
| | Merge Count Penalty Weight | 0.4 | 0.4 |
| | Temporal Distance Penalty Weight | 0.2 | 0.2 |
| Training | Batch Size | 256 | 256 |
| | LR | $1 \times 10^{-5}$ | $1 \times 10^{-5}$ |
| | Epoch | 1 | 1 |
| | Optimizer | AdamW | AdamW |
| | Weight Decay | 0 | 0 |
| | Warmup Ratio | 0.03 | 0.03 |
| | LR Schedule | cosine | cosine |
| | Vision Select Layer | -2 | -2 |
| | GPU Nums | 32 | 32 |

We adopt a five-stage training strategy to systematically train the proposed StreamForest model, aiming to fully exploit its potential for streaming video understanding tasks. In the first three stages, we follow and extend the training paradigm of VideoChat-Flash [33], employing offline training to endow the model with strong capabilities in long-form video comprehension and cross-modal alignment. These stages are designed progressively, covering diverse data scales and task objectives, enabling the model to gradually acquire core competencies such as basic vision-language alignment, long-term temporal modeling, and complex scene reasoning. Detailed training procedures and hyperparameter configurations for these stages are provided in Table 10.

In the fourth and fifth stages, we perform online fine-tuning to enhance the model's ability to process streaming inputs in realistic scenarios. By continuously feeding frame sequences during training, the model learns to retain a fine-grained perception of the current moment while maintaining long-term memory of past events, even under high compression constraints. The full configuration and parameter settings for the online fine-tuning phase are listed in Table 11. These stages are critical for transitioning the model from offline understanding to real-time reasoning, significantly improving its robustness and practical effectiveness in real-world applications.

# E   Full Performances

In the following parts, we present the full results and compare StreamForest with leading proprietary and open-source models. To comprehensively evaluate the effectiveness of StreamForest, we conduct experiments on three online video understanding benchmarks: StreamingBench, OVBench, and OVO-Bench.

## E.1   StreamingBench

Table 12 presents the full evaluation results on StreamingBench, covering 12 real-time video understanding tasks. StreamForest achieves the highest average score (77.26%) among all evaluated models, both open-source and proprietary, while operating efficiently at 1 fps. Notably, StreamForest outperforms leading proprietary MLLMs such as GPT-4o (73.28%) and Gemini 1.5 Pro (75.69%). It also significantly surpasses top open-source offline models such as LLaVA-OneVision (71.12%) and Qwen2.5-VL (73.68%), underscoring its robust multimodal representation and reasoning capabilities. In the online video MLLM category, StreamForest sets a new state-of-the-art, outperforming

Table 12: Full evaluation results of real-time understanding tasks on StreamingBench.

| Method | Size | #Frames | OP | CR | CS | ATP | EU | TR | PR | SU | ACP | CT | ALL |
|---|---|---|---|---|---|---|---|---|---|---|---|---|---|
| Human | - | - | 89.47 | 92.00 | 93.60 | 91.47 | 95.65 | 92.52 | 88.00 | 88.75 | 89.74 | 91.30 | 91.46 |
| *Proprietary MLLMs* | | | | | | | | | | | | | |
| Gemini 1.5 pro [54] | - | 1fps | 79.02 | 80.47 | 83.54 | 79.67 | 80.00 | 84.74 | 77.78 | 64.23 | 71.95 | 48.70 | 75.69 |
| GPT-4o [44] | - | 64 | 77.11 | 80.47 | 83.91 | 76.47 | 70.19 | 83.80 | 66.67 | 62.19 | 69.12 | 49.22 | 73.28 |
| Claude 3.5 Sonnet [1] | - | 20 | 73.33 | 80.47 | 84.09 | 82.02 | 75.39 | 79.53 | 61.11 | 61.79 | 69.32 | 43.09 | 72.44 |
| *Open-source Offline Video MLLMs* | | | | | | | | | | | | | |
| Video-LLaMA2 [10] | 7B | 32 | 55.86 | 55.47 | 57.41 | 58.17 | 52.80 | 43.61 | 39.81 | 42.68 | 45.61 | 35.23 | 49.52 |
| VILA-1.5 [38] | 8B | 14 | 53.68 | 49.22 | 70.98 | 56.86 | 53.42 | 53.89 | 54.63 | 48.78 | 50.14 | 17.62 | 52.32 |
| Video-CCAM [15] | 14B | 96 | 56.40 | 57.81 | 65.30 | 62.75 | 64.60 | 51.40 | 42.59 | 47.97 | 49.58 | 31.61 | 53.96 |
| LongVA [75] | 7B | 128 | 70.03 | 63.28 | 61.20 | 70.92 | 62.73 | 59.50 | 61.11 | 53.66 | 54.67 | 34.72 | 59.96 |
| InternVL2 [9] | 8B | 16 | 68.12 | 60.94 | 69.40 | 77.12 | 67.70 | 62.93 | 59.26 | 53.25 | 54.96 | 56.48 | 63.72 |
| Kangaroo [40] | 7B | 64 | 71.12 | 84.38 | 70.66 | 73.20 | 67.08 | 61.68 | 56.48 | 55.69 | 62.04 | 38.86 | 64.60 |
| LLaVA-NeXT-Video [76] | 32B | 64 | 78.20 | 70.31 | 73.82 | 76.80 | 63.35 | 69.78 | 57.41 | 56.10 | 64.31 | 38.86 | 66.96 |
| MiniCPM-V2.6 [65] | 8B | 32 | 71.93 | 71.09 | 77.92 | 75.82 | 64.60 | 65.73 | 70.37 | 56.10 | 62.32 | 53.37 | 67.44 |
| LLaVA-OneVision [28] | 7B | 32 | 80.38 | 74.22 | 76.03 | 80.72 | 72.67 | 71.65 | 67.59 | 65.45 | 65.72 | 45.08 | 71.12 |
| Qwen2.5-VL [2] | 7B | 1fps | 78.32 | 80.47 | 78.86 | 80.45 | 76.73 | 78.50 | 79.63 | 63.41 | 66.19 | 53.19 | 73.68 |
| *Open-source Online Video MLLMs* | | | | | | | | | | | | | |
| Flash-VStream [72] | 7B | - | 25.89 | 43.57 | 24.91 | 23.87 | 27.33 | 13.08 | 18.52 | 25.20 | 23.87 | 48.70 | 23.23 |
| VideoLLM-online [5] | 8B | 2fps | 39.07 | 40.06 | 34.49 | 31.05 | 45.96 | 32.40 | 31.48 | 34.16 | 34.16 | 42.49 | 27.89 | 35.99 |
| Dispider [47] | 7B | 1fps | 74.92 | 75.53 | 74.10 | 73.08 | 74.44 | 59.92 | 76.14 | 62.91 | 62.16 | 45.80 | 67.63 |
| **StreamForest(Ours)** | 7B | 1fps | 83.11 | 82.81 | 82.65 | 84.26 | 77.50 | 78.19 | 76.85 | 69.11 | 75.64 | 54.40 | 77.26 |

open-source counterparts Dispider (67.63%) by a wide margin. Its consistent accuracy and real-time efficiency demonstrate a strong potential for practical deployment in streaming applications.

## E.2 OVBench

Table 13: Full evaluation results on OVBench.

| Task Name | | FP | | | THV | | | PM | | | SP | | STP | | TP | | | |
|---|---|---|---|---|---|---|---|---|---|---|---|---|---|---|---|---|---|---|---|
| Subset Name | Size | AA | GSP | MP | AP | SV | OP | AR | PR | TR | AL | OP | AT | OT | AS | SL | OES | AVG |
| *Proprietary MLLMs* | | | | | | | | | | | | | | | | | | |
| Gemini-1.5-Flash [54] | - | 71.4 | 53.6 | 21.9 | 56.5 | 60.8 | 40.6 | 36.7 | 47.9 | 62.5 | 32.3 | 37.5 | 87.0 | 50.0 | 83.3 | 22.3 | 46.9 | 50.7 |
| *Open-source Offline Video MLLMs* | | | | | | | | | | | | | | | | | | |
| InternVL2 [9] | 7B | 52.6 | 60.2 | 27.6 | 57.5 | 52.0 | 58.5 | 38.8 | 67.1 | 58.3 | 38.1 | 31.3 | 87.4 | 37.0 | 75.4 | 31.4 | 5.9 | 48.7 |
| InternVL2 [9] | 4B | 57.7 | 57.0 | 14.4 | 59.2 | 49.4 | 60.0 | 30.3 | 61.8 | 46.3 | 30.9 | 20.1 | 83.0 | 32.3 | 70.7 | 29.4 | 3.4 | 44.1 |
| LLaMA-VID [35] | 7B | 43.6 | 50.9 | 19.6 | 64.0 | 47.5 | 46.8 | 29.4 | 48.9 | 51.2 | 31.9 | 11.2 | 75.7 | 24.8 | 59.1 | 26.0 | 40.0 | 41.9 |
| LLaVA-Onevision [28] | 7B | 68.0 | 62.7 | 35.9 | 58.4 | 50.3 | 46.5 | 29.4 | 60.7 | 58.0 | 43.1 | 14.2 | 86.5 | 49.7 | 70.7 | 28.1 | 30.2 | 49.5 |
| LongVA [75] | 7B | 64.1 | 56.5 | 29.5 | 54.9 | 51.9 | 34.8 | 35.3 | 55.6 | 57.7 | 31.6 | 3.4 | 67.4 | 44.7 | 80.0 | 26.7 | 4.0 | 43.6 |
| MiniCPM-V2.6 [65] | 7B | 33.3 | 35.9 | 15.0 | 59.2 | 50.8 | 55.1 | 25.0 | 37.4 | 41.7 | 26.6 | 11.8 | 98.3 | 36.3 | 66.1 | 26.4 | 6.2 | 39.1 |
| Qwen2-VL [55] | 7B | 60.3 | 66.1 | 22.1 | 54.9 | 51.5 | 51.1 | 37.8 | 64.4 | 69.3 | 35.3 | 28.5 | 97.0 | 49.4 | 65.1 | 30.8 | 11.7 | 49.7 |
| LITA [20] | 7B | 19.2 | 24.5 | 19.9 | 40.8 | 48.9 | 24.9 | 3.1 | 27.3 | 6.4 | 6.9 | 14.6 | 35.2 | 23.9 | 27.4 | 0.5 | 3.4 | 20.4 |
| TimeChat [48] | 7B | 7.7 | 15.3 | 18.7 | 20.6 | 15.7 | 11.7 | 9.1 | 14.7 | 9.8 | 7.5 | 19.5 | 13.9 | 10.3 | 9.3 | 10.1 | 10.8 | 12.8 |
| VTimeLLM [19] | 7B | 37.2 | 23.4 | 15.0 | 64.8 | 43.8 | 53.2 | 25.9 | 38.8 | 32.5 | 25.9 | 20.4 | 40.9 | 6.8 | 48.4 | 43.5 | 8.6 | 33.1 |
| *Open-source Online Video MLLMs* | | | | | | | | | | | | | | | | | | |
| VideoLLM-Online [5] | 7B | 0 | 1.8 | 20.9 | 5.2 | 5.9 | 32.6 | 0 | 2.3 | 26.7 | 0.6 | 26.6 | 0.9 | 19.9 | 0.9 | 1.7 | 8.3 | 9.6 |
| MovieChat [52] | 7B | 23.1 | 27.5 | 23.6 | 58.4 | 43.9 | 40.3 | 25.6 | 31.1 | 23.9 | 26.9 | 39.6 | 24.4 | 28.9 | 29.3 | 25.5 | 21.9 | 30.9 |
| Flash-Vstream [72] | 7B | 26.9 | 37.6 | 23.9 | 60.1 | 41.9 | 40.0 | 23.4 | 35.3 | 26.1 | 24.7 | 28.8 | 27.0 | 21.4 | 29.8 | 25.6 | 26.8 | 31.2 |
| Videochat-Online [23] | 4B | 64.1 | 59.7 | 16.6 | 63.1 | 58.3 | 62.8 | 42.2 | 54.4 | 70.6 | 54.1 | 24.8 | 88.7 | 48.5 | 73.0 | 25.9 | 71.7 | 54.9 |
| **StreamForest (Ours)** | 7B | 69.2 | 60.0 | 34.4 | 69.1 | 54.0 | 72.9 | 50.9 | 64.9 | 82.2 | 56.6 | 87.9 | 95.2 | 61.2 | 64.2 | 30.6 | 92.6 | 60.5 |

Table 13 shows the comprehensive results on OVBench, encompassing six diverse task categories (FP, THV, PM, SP, STP, TP). StreamForest achieves the top average score of 60.5%, surpassing all open-source online and offline Video MLLMs. It significantly outperforms other open-source online models, e.g., Videochat-Online (54.9%) and Flash-VStream (31.2%), as well as offline models such as Qwen2-VL (49.7%) and LLaVA-OneVision (49.5%). Compared to Gemini-1.5-Flash (50.7%), StreamForest delivers nearly 10 points higher accuracy on average, affirming its capability to balance real-time efficiency with high performance.

## E.3 OVO-Bench

Table 14 details performance on OVO-Bench, where StreamForest again leads among open-source online video MLLMs with an overall average of 55.57%, outperforming Dispidier-7B (41.78%) and Flash-VStream-7B (33.15%). It excels in critical areas such as real-time visual perception (61.20%), backward tracing (52.02%), and forward active responding (53.49%), showcasing robust temporal

Table 14: Detailed evaluation results on OVO-Bench.

| Model | # Frames | Real-Time Visual Perception | | | | | | | Backward Tracing | | | | Forward Active Responding | | | | Overall Avg. |
|---|---|---|---|---|---|---|---|---|---|---|---|---|---|---|---|---|---|
| | | OCR | ACR | ATR | STU | FPD | OJR | Avg. | EPM | ASI | HLD | Avg. | REC | SSR | CRR | Avg. | |
| Human | - | 93.96 | 92.57 | 94.83 | 92.70 | 91.09 | 94.02 | 93.20 | 92.59 | 93.02 | 91.37 | 92.33 | 95.48 | 89.67 | 93.56 | 92.90 | 92.81 |
| *Proprietary MLLMs* | | | | | | | | | | | | | | | | | |
| Gemini 1.5 Pro [54] | 1fps | 85.91 | 66.97 | 79.31 | 58.43 | 63.37 | 61.96 | 69.32 | 58.59 | 76.35 | 52.64 | 62.54 | 35.53 | 74.24 | 61.67 | 57.15 | 63.00 |
| GPT-4o [44] | 64 | 69.80 | 64.22 | 71.55 | 51.12 | 70.30 | 59.78 | 64.46 | 57.91 | 75.68 | 48.66 | 60.75 | 27.58 | 73.21 | 59.40 | 53.40 | 59.54 |
| *Open-source Offline Video MLLMs* | | | | | | | | | | | | | | | | | |
| LLaVA-Video-7B [76] | 64 | 69.80 | 59.63 | 66.38 | 50.56 | 72.28 | 61.41 | 63.34 | 51.18 | 64.19 | 9.68 | 41.68 | 34.10 | 67.57 | 60.83 | 54.17 | 53.06 |
| LLaVA-OneVision-7B [28] | 64 | 67.11 | 58.72 | 69.83 | 49.44 | 71.29 | 60.33 | 62.79 | 52.53 | 58.78 | 23.66 | 44.99 | 24.79 | 66.93 | 60.83 | 50.85 | 52.88 |
| Qwen2-VL-7B [55] | 64 | 69.13 | 53.21 | 63.79 | 50.56 | 66.34 | 60.87 | 60.65 | 44.44 | 66.89 | 34.41 | 48.58 | 30.09 | 65.66 | 50.83 | 48.86 | 52.70 |
| InternVL2-8B [9] | 64 | 68.46 | 58.72 | 68.97 | 44.94 | 67.33 | 55.98 | 60.73 | 43.10 | 61.49 | 27.41 | 44.00 | 25.79 | 57.55 | 52.92 | 45.42 | 50.05 |
| LongVU-7B [50] | 1fps | 55.70 | 49.54 | 59.48 | 48.31 | 68.32 | 63.04 | 57.40 | 43.10 | 66.22 | 9.14 | 39.49 | 16.62 | 69.00 | 60.00 | 48.54 | 48.48 |
| *Open-source Online Video MLLMs* | | | | | | | | | | | | | | | | | |
| VideoLLM-online-8B [5] | 2fps | 8.05 | 23.85 | 12.07 | 14.04 | 45.54 | 21.20 | 20.79 | 22.22 | 18.80 | 12.18 | 17.73 | - | - | - | - | 12.84 |
| Flash-VStream-7B [72] | 1fps | 25.50 | 32.11 | 29.31 | 33.71 | 29.70 | 28.80 | 29.86 | 36.36 | 33.78 | 5.91 | 25.35 | 5.44 | 67.25 | 60.00 | 44.23 | 33.15 |
| Dispider-7B [47] | 1fps | 57.72 | 49.54 | 62.07 | 44.94 | 61.39 | 51.63 | 54.55 | 48.48 | 55.41 | 4.30 | 36.06 | 18.05 | 37.36 | 48.75 | 34.72 | 41.78 |
| **StreamForest-7B (Ours)** | 1fps | 68.46 | 53.21 | 71.55 | 47.75 | 65.35 | 60.87 | 61.20 | 58.92 | 64.86 | 32.26 | 52.02 | 32.81 | 70.59 | 57.08 | 53.49 | 55.57 |

reasoning across both past and future events. These results position StreamForest as a practical and powerful solution for real-time video-language understanding.

# F Qualitative Comparison

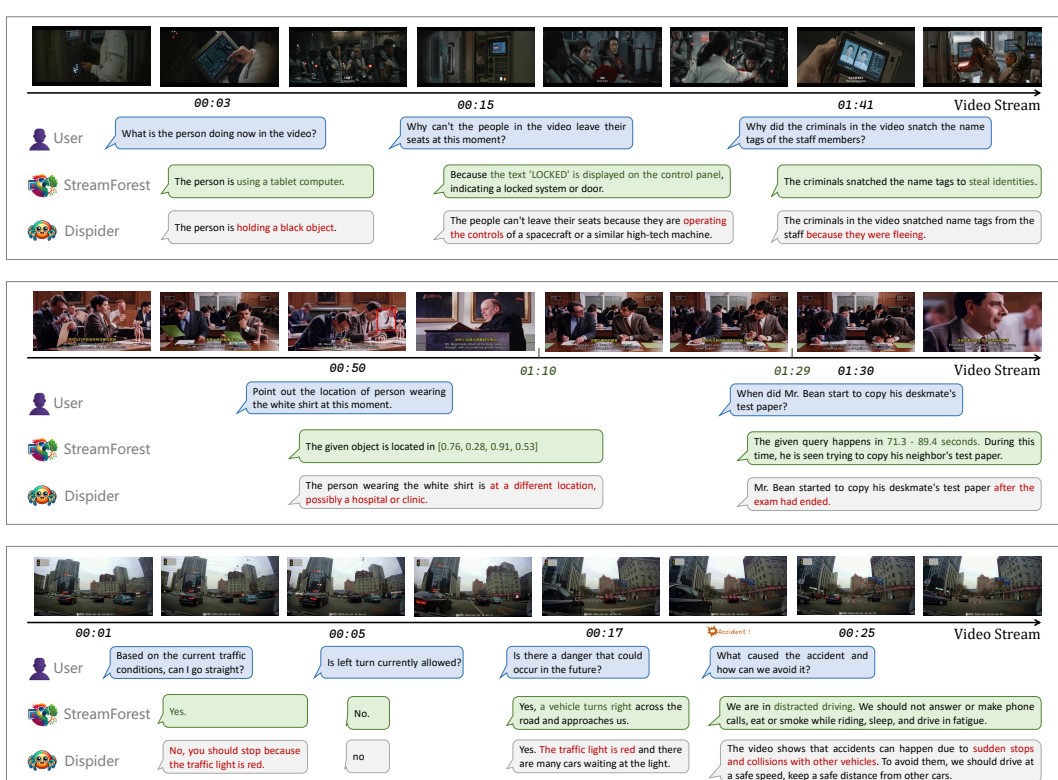

Figure 7: Qualitative comparison between StreamForest and other method.

Figure 7 presents a qualitative comparison between our model and other method. In the top example, StreamForest demonstrates a superior ability to capture fine-grained visual details and maintain persistent memory over time, enabling more coherent and informed inferences. The middle example highlights StreamForest's strong spatiotemporal grounding capabilities, accurately localizing objects and events across space and time. The bottom example illustrates the model's potential in intelligent driving scenarios, where it delivers precise real-time perception and supports future predictions based on both historical and current observations.

# G   More Ablations

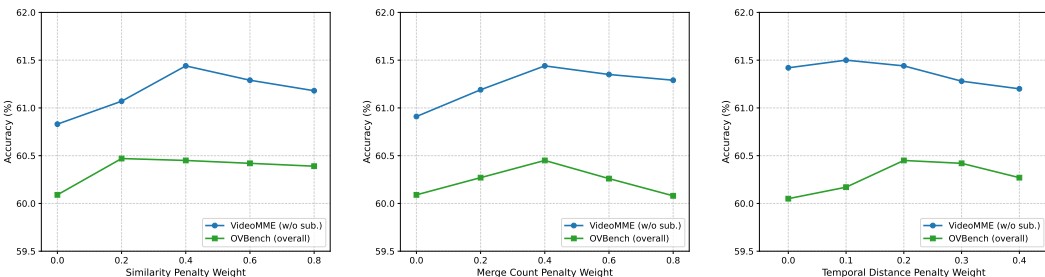

Figure 8: Ablation experiments of three penalty weights.

To evaluate the effect of each penalty on the effectiveness of long-term memory, we conducted ablation studies by varying the weights of the similarity penalty, merge count penalty, and temporal distance penalty. The results are presented in Figure 8, which reports the accuracy of both VideoMME and OVBench under different settings. It demonstrates that a balanced combination of penalty weights is more effective. Specifically, 0.4 for both similarity and merge count penalties, and 0.2 for the temporal distance penalty yield the most effective memory construction. This configuration achieves a favorable trade-off between preserving semantic coherence, maintaining diversity in memory representation, and ensuring reasonable temporal continuity.

# H   Efficiency of Multi-round Inference

We evaluated StreamForest's multi-round response efficiency by streaming a 600-second video to the model at a constant rate of 1 FPS. To isolate processing throughput from text generation latency, the model was constrained to produce a single-token response for each frame. Under this rigorous setting, StreamForest achieved an average processing speed of 9.9 FPS, which is competitive with VideoLLM-Online (12.3 FPS), a model renowned for its real-time capabilities. Crucially, StreamForest delivers this high efficiency without compromising its substantial superiority in reasoning accuracy over VideoLLM-Online. In stark contrast, Qwen2-VL, another model that also prioritizes reasoning accuracy, demonstrated severe performance bottlenecks. Its processing speed dropped below 1 FPS on a video merely two minutes long, and it encountered out-of-memory (OOM) errors on a single A100-80G GPU after processing only 79 frames.

Table 15: Comparison of multi-round inference speed.

| Method | Resolution | FPS |
|---|---|---|
| Qwen2.5-VL | 384 | OOM |
| VideoLLM-Online | 384 | 12.3 |
| StreamForest (1k) | 384 | 9.9 |

# I   Discussions

## I.1   Limitations

Despite the effectiveness of our proposed method, several limitations remain that warrant further investigation. Our approach can only rely on computing inter-frame similarity to determine moments when the model should proactively produce outputs. Specifically, the method identifies local minima in similarity scores to detect transitions. However, this technique primarily captures coarse scene changes and often fails to accurately detect true semantic event boundaries. To address this limitation, one possible solution is to incorporate a lightweight MLLM as an auxiliary reminder module. This module could provide semantic-level guidance to support more precise and context-aware output decisions. These limitations suggest promising directions for future work.

## I.2  Broader Impacts

Our proposed method shows strong potential for real-world streaming video understanding, especially in critical applications like autonomous driving. With domain-specific fine-tuning, it can be adapted to various downstream tasks that require continuous visual processing. As shown in the main text, the model performs well in autonomous driving scenarios, where accurate and timely perception is crucial for safety and decision-making. It can efficiently process live video streams while preserving fine-grained perception and long-term contextual memory. This capability is particularly valuable under limited computational resources, helping improve the reliability and responsiveness of intelligent systems in dynamic environments.

However, as with many vision-language models, potential negative social impacts must also be considered. If deployed without proper safeguards, models may inherit or amplify biases present in training data, leading to unreliable behavior. For instance, performance disparities across different environments or conditions (e.g., weather, lighting, or geographic location) could affect the robustness of StreamForest. To mitigate such risks, we should explore techniques for enhancing interpretability and controllability of streaming video models in safety contexts.

