# OpenReview forum: "StreamForest: Efficient Online Video Understanding with Persistent Event Memory"
_NeurIPS.cc/2025/Conference — NeurIPS 2025 spotlight_

### Official Review · Reviewer_AFDL · 2025-06-30

**Clarity:** 3
**Significance:** 3
**Originality:** 3
**Rating:** 5
**Confidence:** 4

**Summary:**

This paper introduces a novel architecture tailored for real-time video understanding with multimodal large language models (MLLMs). The core of the model consists of two modules: the Fine-grained Spatiotemporal Window, which captures high-resolution visual features for accurate real-time perception, and the Persistent Event Memory Forest, which organizes long-term visual information into adaptive tree structures using penalties based on content similarity, temporal distance, and merge frequency. To support training and evaluation, the authors present OnlineIT, an instruction-tuning dataset for streaming video tasks, and ODV-Bench, a benchmark designed for autonomous driving scenarios. Extensive experiments show that StreamForest outperforms existing online and offline models on multiple online video benchmarks and performs highly competitive with current online and offline models on widely used offline long and short video benchmarks. Moreover, StreamForest maintains high performance under extreme visual token compression, and demonstrates strong robustness, efficiency, and generalization for streaming video understanding.

**Questions:**

* In Table 1, StreamForest performs worse than VideoChat-Online on the Dynamic Target subset. Could you provide any insights into this phenomenon?
* The performance gap between StreamForest and other models on offline short video benchmarks may not be as large as what is reported in Table 2. Recent models, such as Qwen2.5-Omni and InternVL3, have demonstrated strong performance on these datasets.
* How was the base model in Table 5 trained? Was it trained without any data related to online video understanding?

**Ethical Concerns:**

["NO or VERY MINOR ethics concerns only"]

**Final Justification:**

Through the rebuttal, my concerns regarding certain algorithmic implementation details, model training procedures, and the algorithm’s complexity have been addressed. Therefore, I will maintain my initial score.

**Limitations:**

Yes

**Quality:**

3

**Strengths And Weaknesses:**

* Strengths:
    * The two core modules proposed in this work—the Fine-grained Spatiotemporal Window and the Persistent Event Memory Forest—are innovative in design, with ideas that are straightforward yet effective.
    * The experiments in this paper are comprehensive, comparing StreamForest with existing models on both the proposed datasets and established benchmarks, demonstrating the effectiveness of the approach.
    * This paper introduces both a training dataset (OnlineIT) and a testing benchmark (ODV-Bench) for online video understanding. Although these resources have not yet been open-sourced, the authors have committed to releasing them in the future, which is expected to benefit subsequent research in this area.

* Weaknesses:
    * Some details of the proposed algorithm need further clarification. Including a flowchart or block diagram could help present the algorithm more clearly. For example, how should the time of event $t_i$ be computed after two nodes are merged?
    * It would be much better if an experiment comparing the time complexity of different algorithms (e.g. with Pyramid Memory Bank) is included, as maintaining the PEMF involves computations between each pair of root nodes. This potentially costly computation may not meet the efficiency requirements of streaming audio understanding.

Although there is still room for improvement, I believe the overall quality of the paper is good, and I am inclined to recommend its acceptance.

---

> ### Author Rebuttal · Authors · 2025-07-29
>
> We would like to extend our heartfelt gratitude to the reviewer for your insightful comments and valuable feedback. Your questions have significantly contributed to the improvement of our work, and we appreciate the opportunity to address them.
>
> > **W1: Some details of the proposed algorithm need further clarification. Including a flowchart or block diagram could help present the algorithm more clearly. For example, how should the time of event  be computed after two nodes are merged?**
>
>
> Thank you for your valuable suggestion. In the revised version of the manuscript, we will enhance the schematic illustration of StreamForest by adding a flowchart or block diagram to present the algorithm more clearly.
>
> Regarding the computation of event node timestamps: each event node's timestamp is initialized as the average time of the frames it represents (e.g., if an event node spans frames from 10s to 14s, its timestamp is initialized as 12s). When two event nodes are merged, the timestamp of the new node is computed as a token-count-weighted average of the original nodes:
>
> $$t_{\text{new}} = \frac{t_i \cdot n_i + t_j \cdot n_j}{n_i + n_j}$$
>
> where $t_i$, $t_j$ are the timestamps of the original event nodes, and $n_i$, $n_j$ are the numbers of visual tokens contained in each node, respectively.
>
> This weighted scheme prevents timestamp drift during multiple rounds of merging, especially when the merged nodes contain significantly different amounts of visual tokens.
>
>
> > **W2: It would be much better if an experiment comparing the time complexity of different algorithms (e.g. with Pyramid Memory Bank) is included, as maintaining the PEMF involves computations between each pair of root nodes. This potentially costly computation may not meet the efficiency requirements of streaming audio understanding.**
>
> Thank you for the insightful comment. To address this concern, we have added a comparison of PEMF with other memory mechanisms, including the Pyramid Memory Bank [13] and the Similarity Merge strategy [36]. The following table reports the time consumption of each method when processing a 500-second video stream (1 FPS) and producing one effective response at the 500th second.
>
> |Method|Vision Encoding (s)|Memory Update (s)|LLM Inference (s)|**Total (s)**|
> |:-|:-:|:-:|:-:|:-:|
> |Similarity Merge [36]|5.198|0.183|1.388|6.769|
> |Pyramid Memory Bank [13]|5.203|0.451|1.381|7.035|
> |**PEMF (Ours)**|5.218|**0.172**|1.394|6.784|
>
> The results show that vision encoding dominates the overall runtime, processing at 95 FPS, while PEMF’s event consolidation accounts for only 0.172 seconds for 500 frames (2900 FPS). With PEMF integrated, the preprocessing speed for the LLM’s visual inputs decreases by only 3 FPS (5.218+0.172=5.39s, from 95 to 92 FPS), which is negligible compared to the efficiency gains achieved by significantly reducing the total number of visual tokens.
>
> PEMF achieves this efficiency through caching mechanisms: similarity scores, merge counts, and temporal distance states between root nodes are cached in three lists, initialized only once when nodes are created. For each incoming frame, if the visual token budget is exceeded, PEMF performs a single-step merge update. The total penalty for each node pair is directly retrieved from the caches, and after merging, only the similarities with the immediate neighbors need to be updated. Consequently, the overall time complexity grows linearly with the number of frames.
>
> For each individual merge, the dominant cost is computing the similarity matrix between two merged event nodes, which scales as:
>
> $$ T(N,C) \approx O(N^2 C) + O(N \log N) $$
>
> where $N$ is the number of visual tokens contained in the event nodes being merged, $C$ is the feature channel dimension. The $O(N^2 C)$ term arises from the pairwise similarity computations, while subsequent operations such as `max`, `argsort`, and indexing on the matrix add at most $O(N \log N)$ complexity.
>
>
> In contrast, the Pyramid Memory Bank [13] maintains three hierarchical caches with progressively coarser spatial granularity but finer temporal sampling. When the upper-level cache reaches capacity, frames are downsampled and pushed to lower levels, triggering similarity computations with all frames in the lower cache at each replacement step. This additional processing leads to slower memory updates compared to PEMF (0.451s vs. 0.172s for 500 frames).
>
> Overall, PEMF introduces minimal computational burden, scales linearly with stream length, and is demonstrably more efficient than prior memory mechanisms while still enabling persistent, event-level memory retention for streaming video understanding.
>
>
> > **Q1: In Table 1, StreamForest performs worse than VideoChat-Online on the Dynamic Target subset. Could you provide any insights into this phenomenon?**
>
> The Dynamic Target subset primarily focuses on future prediction tasks involving dynamic traffic participants, such as vehicles and pedestrians. These tasks require the model to accurately predict an agent’s next movement or location. For example, determining whether a vehicle will change lanes or continue straight, or estimating spatial relations such as approaching or receding distance.
>
> To better simulate real-world traffic, the benchmark includes challenging cases where agents behave unpredictably or violate traffic rules. For instance, as shown in the [AP] example in Figure 1 (in the supplementary material), a bus is driving in a right-turn-only lane. While the correct future action is (D) Moving to the left lane (as the bus illegally crosses a solid line), our model, which tends to rely on explicit cues such as road signs and lane markings, was misled by the right-turn marking and incorrectly predicted (B) Turning right. This is reflected in the performance drop on the AP task from 73.4 to 57.0 after fine-tuning with OnlineIT.drive, which does not yet include sufficient data for such counter-intuitive behaviors.
>
> Moreover, VideoChat-Online benefits slightly on this subset because its original training data is biased toward short-term dynamic tasks, which overlap with the demands of these prediction-oriented evaluations. In contrast, StreamForest’s training strategy was designed to balance performance across eight diverse benchmarks while controlling training costs, leading to some trade-offs in short-term dynamic prediction coverage. To validate this explanation, we conducted a supplementary experiment by post-training StreamForest with the full VideoChat-Online dataset. As shown below, StreamForest then outperformed VideoChat-Online on the Dynamic Target subset:
>
> |Method|AP|LP|DP|Dynamic Target (Avg)|
> |:-|:-:|:-:|:-:|:-:|
> |VideoChat-Online|72.6|89.3|48.1|65.6|
> |StreamForest|73.4|81.2|47.9|63.3|
> |StreamForest (FT-drive)|57.0|96.8|46.2|63.3|
> |StreamForest (post-trained on full VideoChat-Online data)|74.3|90.1|48.6|**67.8**|
>
> In future work, we plan to expand our dataset with more high-quality, behavior-diverse, future-prediction instruction-tuning data to further strengthen the model’s capability on dynamic target tasks.
>
>
> > **Q2: The performance gap between StreamForest and other models on offline short video benchmarks may not be as large as what is reported in Table 2. Recent models, such as Qwen2.5-Omni and InternVL3, have demonstrated strong performance on these datasets.**
>
> Thank you for the suggestion. In the revised version of the manuscript, we will include comparisons with more recent models such as Qwen2.5-Omni and InternVL3, which have reported competitive or superior results on short-video evaluation benchmarks. These results were part of contemporaneous work that had not yet undergone peer review at the time of our submission. Therefore, Table 2 in the initial manuscript focused on widely used open-source baselines from prior studies.
>
> It is also important to note that our primary contribution is not to surpass all offline short-video MLLMs, which is particularly challenging under streaming video settings, but rather to enable robust streaming video understanding through our Persistent Event Memory Forest and Fine-grained Spatiotemporal Window . These innovations specifically target the unique challenges of streaming scenarios, such as long-term memory retention and real-time perception, which are not directly addressed by most offline models.
>
>
> > **Q3: How was the base model in Table 5 trained? Was it trained without any data related to online video understanding?**
>
> The base model in Table 5 was primarily trained on offline video data sampled from VideoChat-Flash and LLaVA-Video, along with image data from LLaVA-OneVision. Additionally, to ensure the model could adapt to the response patterns required for online video understanding, we incorporated a small subset of raw data drawn from VideoChat-Online (including Charades-STA, AVA, LaSOT, COIN, and ActivityNet-Captions, totaling approximately 32K samples). To further enhance the model’s fine-grained perception capability, we also included a portion of samples  from open-source image-based spatial grounding datasets RefCOCO (approximately 30K).
>
> Although the base dataset shares some video data sources with OnlineIT.general, all data used for the base model was uncleaned, original data. In contrast, all cleaned and re-annotated data are fully contained within OnlineIT.general.

---

> > ### Comment · Reviewer_AFDL · 2025-08-05
> > **follow-up**
> >
> > Thank you for the detailed explanation. My concerns have been addressed.

---

> > > ### Author Response · Authors · 2025-08-06
> > >
> > > Dear Reviewer AFDL,
> > >
> > > We sincerely thank you for your constructive and detailed feedback throughout the review process. Your comments helped us refine key aspects of our method, particularly in terms of algorithmic clarity and computational analysis. We are glad to know that our responses addressed your concerns.

---

### Official Review · Reviewer_gQt1 · 2025-07-01

**Clarity:** 2
**Significance:** 2
**Originality:** 2
**Rating:** 4
**Confidence:** 3

**Summary:**

The paper proposes StreamForest for streaming video understanding. The key contribution in the modeling part is Persistent Event Memory Forest (PEMF), a memory mechanism that adaptively merges video frames into event-level tree structures. Additionally, the authors propose a Fine-grained Spatiotemporal Window to enhance real-time perception by capturing detailed short-term visual cues. The paper also introduces an instruction-tuning dataset OnlineIT, and ODV-Bench, a new benchmark for evaluating streaming video understanding in autonomous driving scenarios. Through extensive experiments, the authors demonstrate that StreamForest achieves state-of-the-art performance on multiple benchmarks.

**Questions:**

My critical concern lies in the technical contribution and organization of the paper. The author could highlight the main contribution and reformat the paper. If the dataset/benchmark is a critical contribution, it should be verfied across different mllm archs.

**Ethical Concerns:**

["NO or VERY MINOR ethics concerns only"]

**Final Justification:**

After rebuttal, I think most of my concerns are addressed. Thus I will give a borderline accept score.

**Limitations:**

The author has discussed the limitations.

**Quality:**

2

**Strengths And Weaknesses:**

For strengths:

1. The paper proposes a new method along with a new instruction tuning dataset along a new benchmark, which can benefit the whole community.

2. The proposed method has some ablation studies to verify its effectiveness.

However, I have the following concerns for the paper:

1. The organization of the paper is somewhat chaotic. Since the author proposes a new model, a new instruction tuning dataset, and a new benchmark. It sounds like those three components are independent, so why should they be composed in one paper? When I read the paper, it is more like an A+B+C paper without a main topic. The author should reconsider the motivation and reformat the paper.
2. Does the comparison fair? Since the author proposes a new model, a new instruction tuning dataset, and a new benchmark, as well as results on those benchmarks. Though the author provides some ablation studies in Table 3~5. But I noticed that with the baseline dataset, the proposed method only has comparable performance or even lower (like 54.9 vs 53.5), so I think the performance gain is mainly from the dataset part. So maybe the paper is more suitable for the dataset and benchmark track? I think for the main track, the technical novelty may not be sufficient.

---

> ### Author Rebuttal · Authors · 2025-07-29
>
> We sincerely appreciate your careful evaluation of our work and present our detailed responses as follows.
>
> >**W1: The organization of the paper is somewhat chaotic. Since the author proposes a new model, a new instruction tuning dataset, and a new benchmark. It sounds like those three components are independent, so why should they be composed in one paper? When I read the paper, it is more like an A+B+C paper without a main topic. The author should reconsider the motivation and reformat the paper.**
>
> The three contributions of our work are not independent components but complementary elements designed to address three fundamental gaps in streaming video understanding: model architecture, training data, and evaluation. **All three share a unified motivation: To enhance the performance of StreamForest on streaming video understanding tasks and to provide a rigorous, application-driven evaluation.**
>
> **At the core of our work is the StreamForest architecture, which directly addresses two key shortcomings of existing online video MLLMs:** The absence of scalable memory and the limitation in fine-grained spatiotemporal reasoning. StreamForest introduces (1) the Persistent Event Memory Forest (PEMF) for adaptive long-term event storage, and (2) the Fine-grained Spatiotemporal Window (FSTW) for precise real-time perception, enabling efficient handling of long-horizon, dynamic video streams.
>
> **The OnlineIT dataset is designed to unlock the full potential of StreamForest.** Its construction explicitly targets three capabilities: (1) persistent memory of past events (aligned with PEMF), (2) fine-grained perception of the current spatiotemporal context (aligned with FSTW), and (3) prediction of future states (where PEMF and FSTW jointly contribute). The dataset and model architecture are thus tightly coupled, and their synergy leads to substantial gains in streaming video tasks.
>
> **The ODV-Bench evaluates StreamForest in real-world downstream scenarios that existing benchmarks fail to capture.** Unlike prior streaming benchmarks, ODV-Bench emphasizes rapidly evolving, dynamic environments with complex object motion, multi-agent interactions, and high-stakes prediction tasks. Testing StreamForest on ODV-Bench highlights how our approach outperforms existing methods under the very conditions it was designed for, thereby showcasing its practical value.
>
> **Together, these three components form a cohesive framework rather than isolated contributions:** StreamForest (the architecture) delivers the core capability, OnlineIT equips it with aligned training data, and ODV-Bench validates its advantages in realistic streaming video applications.
>
>
> >**W2: Does the comparison fair? Since the author proposes a new model, a new instruction tuning dataset, and a new benchmark, as well as results on those benchmarks. Though the author provides some ablation studies in Table 3~5. But I noticed that with the baseline dataset, the proposed method only has comparable performance or even lower (like 54.9 vs 53.5), so I think the performance gain is mainly from the dataset part. So maybe the paper is more suitable for the dataset and benchmark track? I think for the main track, the technical novelty may not be sufficient.**
>
>
> **On the fairness of the comparison:** In addition to evaluating StreamForest on our proposed ODV-Bench, we also validate its effectiveness on seven widely recognized, high-quality open-source benchmarks (Table 2 in the paper). All evaluations strictly follow each benchmark’s official protocols, ensuring that our comparisons fairly reflect the capabilities of our method relative to existing approaches.
>
> **On the validity of the ablations:** Table 3 shows that, under the same instruction-tuning data and visual token budgets, replacing PEMF with existing methods results in a 1.6–3.4% drop in average accuracy on OVBench, OVO-Bench, and MLVU. The drop is most severe on MLVU, which requires robust long-term memory retention. Similarly, Table 4 demonstrates that removing either PEMF or FSTW leads to significant performance degradation, confirming the unique contributions of both components in StreamForest’s architecture.
>
> To further validate PEMF’s contribution, we extend Figure 4 by comparing it with the Similarity Merge [36] under different token budgets. The results are summarized below (accuracy with retention percentage relative to each method’s 8k-token setting):
>
> |Visual Token Budget|Method|OVO-Bench|VideoMME|MLVU|
> |:-|:-|:-:|:-:|:-:|
> |**1k**|Similarity Merge [36]|52.7 (98.7%)|56.2 (93.5%)|60.4 (88.8%)|
> | |**PEMF (Ours)**|**55.3 (99.5%)**|**58.5 (95.3%)**|**62.9 (89.9%)**|
> |**2k**|Similarity Merge [36]|52.8 (98.9%)|58.1 (96.7%)|63.5 (93.3%)|
> | |**PEMF (Ours)**|**55.7 (100.2%)**|**60.6 (98.7%)**|**65.9 (94.1%)**|
> |**4k**|Similarity Merge [36]|53.1 (99.4%)|59.8 (99.5%)|65.2 (95.8%)|
> | |**PEMF (Ours)**|**55.8 (100.4%)**|**61.4 (100%)**|**67.6 (96.5%)**|
> |**8k (default)**|Similarity Merge [36]|53.4 (100%)|60.1 (100%)|68.0 (100%)|
> | |**PEMF (Ours)**|**55.6 (100%)**|**61.4 (100%)**|**70.0 (100%)**|
>
> PEMF consistently outperforms Similarity Merge not only in absolute accuracy but also in performance retention under extreme compression, most notably on VideoMME (e.g., at 1k tokens: +2.3% absolute accuracy, +1.8% retention). These results confirm that StreamForest’s performance improvements arise primarily from its architecture’s robust event-level memory design, not merely from additional data.
>
> **On the role of instruction-tuning data:** **Even when fine-tuned solely on the base dataset, StreamForest surpasses existing state-of-the-art streaming models on most benchmarks.** For example, it achieves 53.5% vs. 41.8% on OVO-Bench (vs. Dispider, +11.7%) and 56.3% vs. 55.7% on ODV-Bench (vs. VideoChat-Online, +0.6%). Only on OVBench does it slightly trail VideoChat-Online (53.9% vs. 54.9%, -1.0%).
>
> **It is also important to clarify that the “base” dataset used in Table 5 is not directly comparable to the instruction-tuning datasets used by other models.** Specifically, VideoChat-Online’s online instruction-tuning data (which OVBench relies on) was not fully included in the base set. To isolate the effect of instruction data, we fine-tuned StreamForest on the base dataset plus the full VideoChat-Online instructions, and report the results below:
>
> |Training Data|ODV-Bench (Avg)|OVBench (Avg)|OVO-Bench (Overall)|
> |:-|:-:|:-:|:-:|
> |base|56.3|53.9|53.5|
> |base + VideoChat-Online (full)|57.4|56.1|54.0|
> |base + O.general|60.5|60.5|55.6|
> |base + O.general + O.drive|67.3|60.5|54.5|
>
> After incorporating the same instruction-tuning data used by VideoChat-Online, StreamForest surpasses it on OVBench (56.1% vs. 54.9%), confirming that the performance gains of StreamForest cannot be attributed to OnlineIT data alone.
>
> **In summary, StreamForest’s improvements stem from its novel architecture, particularly PEMF’s event-level memory consolidation and FSTW’s localized perception, with instruction-tuning data serving as a complementary factor rather than the primary driver. Therefore, the method fits the main technical track, as it introduces substantive architectural innovations validated by extensive ablations and cross-benchmark testing.**
>
>
> >**Q1: My critical concern lies in the technical contribution and organization of the paper. The author could highlight the main contribution and reformat the paper. If the dataset/benchmark is a critical contribution, it should be verfied across different mllm archs.**
>
> Thank you for the feedback. **The primary contribution of our work is StreamForest, a novel architecture that integrates the PEMF and FSTW to jointly address persistent memory and real-time perception in streaming video understanding.** These challenges cannot be solved by scaling data alone, as existing MLLMs still suffer from uncontrolled memory growth, inefficient long-term reasoning, and degraded temporal coherence when handling long-horizon, dynamic inputs. The inclusion of the OnlineIT dataset and ODV-Bench benchmark is intended to unlock the full potential of StreamForest and to provide rigorous, application-driven evaluation, rather than to serve as independent contributions.
>
> **The organization of the paper already reflects this prioritization.** The StreamForest architecture is clearly positioned as the main contribution in the title, abstract, and throughout the manuscript. In the methodology section, two pages (L98–L173) are dedicated to the architecture’s technical innovations, while the OnlineIT dataset and ODV-Bench benchmark together account for only ~1.5 pages (L174–L238). Similarly, the experiments (Tables 1–4, Figures 3–4) are designed primarily to analyze and validate StreamForest, with only limited space (Table 1, Table 5) devoted to the benchmark and dataset.
>
> **In terms of technical novelty, StreamForest introduces two innovations that distinguish it from prior Video MLLMs:**
>
> - **Event-level persistent memory with adaptive token control:** PEMF consolidates historical visual tokens into a structured forest of event nodes, guided by multi-criteria penalties on similarity, merge counts, and temporal distance. This mechanism prevents uncontrolled token growth, preserves semantically salient events, and maintains coherent memory over thousands of frames, all of which are absent in existing approaches.
>
> - **Localized, fine-grained temporal perception:** FSTW dynamically allocates high-resolution windows near the current query timestamp, ensuring the model retains rich local context without overwhelming the LLM with redundant tokens.
>
> In the revised manuscript, we will further refine the organization to highlight StreamForest as the core technical contribution, make this hierarchy explicit in the introduction, and incorporate the new ablation results presented during rebuttal to more comprehensively substantiate the effectiveness of the proposed architecture.

---

> ### Author Response · Authors · 2025-08-05
> **Kind Request to Reply Our Rebuttal**
>
> Dear Reviewer gQt1,
>
> Thank you once again for your feedback, which has significantly contributed to the improvement of our work.
>
> We would like to kindly remind you that we have responded to the concerns you previously raised. As the discussion period is drawing to a close, we would sincerely appreciate it if you could let us know whether you have any remaining questions or concerns. Please feel free to contact us at your earliest convenience so that we can address them in a timely manner.
>
> Thank you for your time and consideration.

---

### Official Review · Reviewer_WUtt · 2025-07-02

**Clarity:** 3
**Significance:** 3
**Originality:** 2
**Rating:** 5
**Confidence:** 5

**Summary:**

The paper proposes a streaming VideoLLM consisting of a persistent event memory forest (PEMF), which introduces three penalty functions in content similarity, merge frequency, and temporal distance. PEMF enables storing very long past context at reduced token count and hence reduced memory. In contrast, short-term memory is maintained at high resolution for capturing fine-grained details. A new instruction-tuning dataset, OnlineIT, is also proposed along with a new streaming video benchmark, ODV-Bench. The proposed model, along with OnlineIT, achieves SOTA on multiple benchmarks.

**Questions:**

I will consider the authors' responses to both questions while deciding my final rating.
- **Q1.** I am not fully convinced by Fig. 4, as it should ideally include comparisons with other merging methods like [36]. Otherwise, the claim in L317–L318 and L20–L21 remains unsubstantiated. Is this a general trend across video benchmarks and different models, or is the proposed model genuinely helping with token efficiency?
- **Q2.** While the paper is framed around streaming video understanding, a key aspect in real-time, low-latency performance is not discussed. In fact, PEMF is likely to increase latency due to its multiple online operations, which may hinder techniques like prefilling and KV cache usage. From Fig. 5, it appears the model runs at around 1 fps on average, which is significantly slower than existing streaming models like VideoLLM-online [4] (~13.5 fps). Can the authors clarify their motivation for presenting this as a streaming approach despite the latency constraints?

The following question will not affect my rating, but a response would be appreciated.
- **Q3.** Why is the proposed benchmark focused on autonomous driving scenarios rather than something else like daily activities (e.g., Ego4D), where videos are naturally captured online and streaming understanding is equally or likely more important? Is there a specific motivation behind this choice?

**Ethical Concerns:**

["NO or VERY MINOR ethics concerns only"]

**Final Justification:**

I thank the authors for their detailed responses to my queries and concerns in the rebuttal.

The rebuttal has improved my understanding of the work and my overall appreciation of the paper, and I am recommending an Accept. I strongly urge the authors to revise the writing throughout the manuscript. For example, the response to Q3 in the rebuttal is clear and well-written and would be very helpful if included in the main paper. Also, I encourage the authors to include these efficiency statistics in the main paper, which will further strengthen the work.

**Limitations:**

yes

**Paper Formatting Concerns:**

I did not notice any major formatting issues.

**Quality:**

3

**Strengths And Weaknesses:**

**Strengths:**
- **S1.** The proposed long-short-term memory approach is a generalization of existing efforts under one end-to-end trainable architecture. The simple and elegant design is appealing.
- **S2.** StreamForest achieves state-of-the-art performance on multiple benchmarks including streaming, long-term, and short-term ones.
- **S3.** The model ablation is well-conducted, and the supplementary material is comprehensive with dataset details and model hyperparameters for reproducibility.

**Weaknesses:**
- **W1.** Upon closer inspection, it seems that a significant portion of the performance improvement may stem from the larger/better instruction-tuning dataset, rather than the proposed model itself. In Table 2, comparable online VideoLLMs such as [4, 46, 50], especially MovieChat [36], are trained on significantly less data. In the Table 3 ablation, it appears that using the same instruction-tuning data, the performance difference between [36] and the proposed method is relatively small. Improvements primarily driven by a better instruction-tuning dataset are less appealing and not the main focus of the paper.
- **W2.** The discussion section in the experiments is underdeveloped and does not adequately reflect the observed performance. It is written in a rather generic manner, and connections to specific results are missing. For example, in L294–L297, the authors dismiss similarity-based merging due to “severe spatiotemporal irregularities,” yet Table 3 shows that the accuracy of Similarity Merge is quite comparable to PEMF. Similarly, L307–L309 is also broadly written and not well-grounded in the performance observed in Table 4. This defeats the purpose of ablations.

---

> ### Author Rebuttal · Authors · 2025-07-28
>
> We would like to express our sincere gratitude for your review. Your suggestions will greatly help us improve our paper.
>
> >**W1: Upon closer inspection, it seems that a significant portion of the performance improvement may stem from the larger/better instruction-tuning dataset, rather than the proposed model itself. In Table 2, comparable online VideoLLMs such as [4, 46, 50], especially MovieChat [36], are trained on significantly less data. In the Table 3 ablation, it appears that using the same instruction-tuning data, the performance difference between [36] and the proposed method is relatively small. Improvements primarily driven by a better instruction-tuning dataset are less appealing and not the main focus of the paper.**
>
> While the OnlineIT dataset contributes positively to the overall performance of StreamForest, the primary source of improvement does not solely lie in the dataset’s scale but rather in our architectural innovations. In particular, the Persistent Event Memory Forest (PEMF) enables the model to maintain long-term, event-level persistent memory, which is a capability that cannot be achieved by instruction-tuning data alone.
>
> As shown in Table 3 in the paper, even when trained on the same instruction-tuning dataset, StreamForest still achieves notable performance gains compared with Similarity Merge [36], including a +2.2% improvement on OVO-Bench and +2.0% on MLVU. These results confirm that persistent memory architecture, rather than data scale alone, is crucial for effectively handling long-horizon streaming video inputs. Furthermore, the additional experiments we present in response to Q1 (comparing PEMF with other merging strategies under varying token budgets) reinforce this conclusion by demonstrating PEMF’s robustness and superior token efficiency.
>
> >**W2: The discussion section in the experiments is underdeveloped and does not adequately reflect the observed performance. It is written in a rather generic manner, and connections to specific results are missing. For example, in L294–L297, the authors dismiss similarity-based merging due to “severe spatiotemporal irregularities,” yet Table 3 shows that the accuracy of Similarity Merge is quite comparable to PEMF. Similarly, L307–L309 is also broadly written and not well-grounded in the performance observed in Table 4. This defeats the purpose of ablations.**
>
> Thank you for pointing this out. We agree that the discussion in the ablation section should be more explicitly tied to the quantitative results, so that task-specific performance gaps are clearly reflected. In the revised manuscript, we will refine this section to ground the analysis in the observed data.
>
> For example, the modifications to lines L294–L297 are as follows: OVBench primarily emphasizes short-term, fine-grained spatiotemporal perception. Similarity Merge achieves performance comparable to our PEMF on OVBench (60.3% vs. 60.5%). However, its limitations become clear in tasks that require persistent memory and long-horizon reasoning. On OVO-Bench, PEMF outperforms Similarity Merge by +2.2%, and on MLVU by +2.0%. This is because similarity-based merging may over-merge frames within local video segments, potentially leading to spatiotemporal irregularities and the loss of local event-level representations.
>
> >**Q1: I am not fully convinced by Fig. 4, as it should ideally include comparisons with other merging methods like [36]. Otherwise, the claim in L317–L318 and L20–L21 remains unsubstantiated. Is this a general trend across video benchmarks and different models, or is the proposed model genuinely helping with token efficiency?**
>
> Thank you for the detailed feedback. To further substantiate our claim regarding token efficiency and validate the robustness of our proposed Persistent Event Memory Forest (PEMF), we extend Figure 4 by including a direct comparison with the Similarity Merge [36] under varying visual token budgets. The following table summarizes the results at different visual token budgets. For each setting, we report both the absolute accuracy and the percentage of performance retained relative to default(8k).
>
> |Visual Token Budget|Method|OVO-Bench|VideoMME|MLVU|
> |:-|:-|:-:|:-:|:-:|
> |**1k**|Similarity Merge [36]|52.7 (98.7%)|56.2 (93.5%)|60.4 (88.8%)|
> | |**PEMF (Ours)**|**55.3 (99.5%)**|**58.5 (95.3%)**|**62.9 (89.9%)**|
> |**2k**|Similarity Merge [36]|52.8 (98.9%)|58.1 (96.7%)|63.5 (93.3%)|
> | |**PEMF (Ours)**|**55.7 (100.2%)**|**60.6 (98.7%)**|**65.9 (94.1%)**|
> |**4k**|Similarity Merge [36]|53.1 (99.4%)|59.8 (99.5%)|65.2 (95.8%)|
> | |**PEMF (Ours)**|**55.8 (100.4%)**|**61.4 (100%)**|**67.6 (96.5%)**|
> |**8k (default)**|Similarity Merge [36]|53.4 (100%)|60.1 (100%)|68.0 (100%)|
> | |**PEMF (Ours)**|**55.6 (100%)**|**61.4 (100%)**|**70.0 (100%)**|
>
> **The results demonstrate that PEMF has two core benefits:**
> - **Superior absolute performance:** PEMF outperforms Similarity Merge [36] across all token budgets, with an average absolute performance improvement of 2%-3%.
> - **Stronger resilience to extreme compression:** PEMF retains a higher fraction of its full-budget performance under severe compression (at 1k tokens, +1.8% retention on VideoMME).
>
> The above experimental results confirm that the performance gains are not merely a general trend across benchmarks but stem from the intrinsic design of PEMF, which adaptively consolidates event-level memory and preserves semantically salient information. This mechanism enables StreamForest to achieve both competitive accuracy and high token efficiency under stringent resource constraints.
>
> >**Q2: While the paper is framed around streaming video understanding, a key aspect in real-time, low-latency performance is not discussed. In fact, PEMF is likely to increase latency due to its multiple online operations, which may hinder techniques like prefilling and KV cache usage. From Fig. 5, it appears the model runs at around 1 fps on average, which is significantly slower than existing streaming models like VideoLLM-online [4] (~13.5 fps). Can the authors clarify their motivation for presenting this as a streaming approach despite the latency constraints?**
>
> It is not accurate to conclude from Figure 5 that our method operates at only 1 FPS based on the reported inference latency (~1s). The reported latency reflects the average time required to generate a complete response to a user query (i.e., from receiving a question to producing a full answer). In contrast, VideoLLM-Online [4] counts the non-informative silent tokens as part of its FPS rate. Its actual generation of meaningful responses is much sparser, as the model frequently outputs only [EOS] tokens to indicate silence when no answer is needed.
>
> In practical applications, the effective FPS of a streaming model is heavily influenced by the frequency of questions and responses. Therefore, we report the latency per valid response as a more reliable measure of system speed. As shown in the table below, our approach does not suffer from a computational efficiency disadvantage compared to existing methods.
>
> |Method|Inference Latency (s)|
> |:-|:-:|
> |VideoLLM-Online [4]|0.846|
> |Dispider [32]|1.539|
> |StreamForest (1k)|0.867|
> |StreamForest (8k)|1.489|
>
> The computational cost of updating PEMF is negligible compared to that of the vision encoder and LLM. We have added a comparison of PEMF with other memory mechanisms, including the Pyramid Memory Bank [13] and the Similarity Merge strategy [36]. The following table reports the time consumption of each method when processing a 500-second video stream (1 FPS) and producing one effective response at the 500th second.
>
> |Method|Vision Encoding (s)|Memory Update (s)|LLM Inference (s)|**Total (s)**|
> |:-|:-:|:-:|:-:|:-:|
> |Similarity Merge [36]|5.198|0.183|1.388|6.769|
> |Pyramid Memory Bank [13]|5.203|0.451|1.381|7.035|
> |**PEMF (Ours)**|5.218|**0.172**|1.394|6.784|
>
> The results show that vision encoding dominates the overall runtime, processing at 95 FPS, while PEMF’s event consolidation accounts for only 0.172 seconds for 500 frames (2900 FPS). With PEMF integrated, the preprocessing speed for the LLM’s visual inputs decreases by only 3 FPS (5.218+0.172=5.39s, from 95 to 92 FPS), which is negligible compared to the efficiency gains achieved by significantly reducing the total number of visual tokens.
>
> Theoretically, PEMF is also compatible with KV-Cache. Each event-level merge modifies only a small subset of visual tokens, so KV-Cache recomputation and prefilling are required only for those regions (There is no need to recompute the KV-Cache for all subsequent tokens beyond the modified region, since they were originally computed with a more complete set of preceding visual information). With stable frame inputs, each event consolidation compresses visual tokens in the affected interval by ~50%, meaning the number of recomputed KV-Cache tokens is roughly equal to the number of new tokens added to PEMF. As a result, KV-Cache update cost is roughly doubled compared to the no-memory case.
>
> >**Q3: Why is the proposed benchmark focused on autonomous driving scenarios rather than something else like daily activities (e.g., Ego4D), where videos are naturally captured online and streaming understanding is equally or likely more important? Is there a specific motivation behind this choice?**
>
> Previous streaming benchmarks, such as OVO-Bench, already incorporate Ego4D videos of daily activities. These benchmarks primarily evaluate MLLMs’ ability to perceive static scenes and narrate human–environment interactions in a stepwise manner.
>
> In contrast, autonomous driving presents dynamic, high-stakes environments with rapidly changing scenes, complex multi-agent interactions (vehicles, pedestrians, and traffic signals), and demanding prediction tasks such as risk assessment and motion planning. These scenarios require models to balance long-term event memory with fine-grained short-term perception to avoid accidents and make timely decisions.

---

> > ### Comment · Reviewer_WUtt · 2025-08-06
> > **Response to rebuttal**
> >
> > I thank the authors for their detailed responses to my queries and concerns in the rebuttal.
> >
> > The rebuttal has improved my understanding of the work and my overall appreciation of the paper, and I am now leaning toward an Accept. I strongly urge the authors to revise the writing throughout the manuscript. For example, the response to Q3 in the rebuttal is clear and well-written and would be very helpful if included in the main paper.
> >
> > I would also like to clarify my question regarding time consumption and efficiency. The table comparing [36, 13] with PEMF addresses runtime at the end of a 500-second video, but my question was about the model’s ability to operate in a streaming context, specifically, its capacity to memorize past content and answer multiple questions continuously as new segments are processed, where minimizing re-computation is essential. Although this aspect was not fully addressed, the additional performance comparisons with VideoLLM-online and Dispider in the rebuttal paints a good picture that the model is efficient compared to SOTA baselines. I am not requesting further clarification, but I again encourage the authors to include these efficiency statistics in the main paper, as this would strengthen the paper and broaden its appeal.

---

> > > ### Author Response · Authors · 2025-08-06
> > >
> > > Dear Reviewer WUtt,
> > >
> > > We sincerely thank you for your thoughtful and in-depth review. Your constructive engagement throughout the review process has significantly improved the clarity and completeness of our work. We are pleased to hear that our rebuttal deepened your understanding of the paper and contributed to your improved overall assessment. This is very encouraging for us.
> > >
> > > We particularly appreciate your feedback on the clarity of writing and the motivation behind the benchmark design. In the revised version, we will incorporate the clearer and more detailed explanations, especially those related to Question 3, to ensure that our contributions are more accessible and well motivated.
> > >
> > > We also thank you for emphasizing the importance of time efficiency. We will include the relevant efficiency statistics discussed in Question 2 in the main text. While our rebuttal focused on single-turn latency and total runtime, we fully agree that maintaining real-time performance under multiple continual queries is a crucial aspect of streaming video understanding. In the camera-ready version, we will extend our discussion to better reflect the inference efficiency of our model in continuous streaming scenarios. Additionally, we plan to explore incorporating techniques such as KV-Cache to further reduce redundant computation in future work.
> > >
> > > Once again, thank you for your valuable suggestions and for recognizing the improvements we have made. Your feedback has played an important role in strengthening both the substance and the presentation of our work.

---

### Official Review · Reviewer_nvw1 · 2025-07-02

**Clarity:** 4
**Significance:** 3
**Originality:** 3
**Rating:** 5
**Confidence:** 2

**Summary:**

This paper introduces StreamForest, a novel architecture for streaming video understanding that incorporates a Persistent Event Memory Forest to efficiently manage historical visual information. Memories are merged based on temporal distance, content similarity, and merge count penalties, allowing the model to retain long-term context under constrained resources. In addition, the authors present OnlineIT, a new instruction-tuning dataset tailored for streaming video tasks, and ODV-Bench, a benchmark focused on real-time autonomous driving scenarios. The proposed method shows substantial improvements over state-of-the-art MLLMs, especially in constrained-token regimes.

**Questions:**

The ODV-Bench could be described in more detail. For example, "task examples" are mentioned in Figure 3(c), the paper does not clearly define a taxonomy of question types (e.g., object recognition, temporal reasoning, causal inference, prediction). I think such a taxonomy would be useful for interpreting performance in this benchmark.

Quality control in the construction of the benchmark seems under-reported. "multiple rounds of manual review," could be clarified with quantitative information regarding inter-annotator agreement, human accuracy baselines, variability as a function of taxonomy.

Joint ablations across e.g. FSTW and PEMF could be included to identify synergistic benefits.

**Ethical Concerns:**

["NO or VERY MINOR ethics concerns only"]

**Final Justification:**

I've reflected on the authors' proposed changes which will improve the quality and clarity of this paper imo thus Ive increased these scores.

**Limitations:**

The limitations are only discussed in the SM, I think limitations should also be summarized in the main ms.

**Paper Formatting Concerns:**

yes

**Quality:**

4

**Strengths And Weaknesses:**

This work tackles a challenging and impactful problem that is of current interest, namely the efficient long-term understanding of streaming video. While the use of memory structures for summarization is not entirely novel, the proposed integration into MLLMs and the accompanying benchmarks demonstrate a meaningful advance. The results suggest real practical value, especially under resource constraints.

The paper is technically solid, and the proposed components are clearly motivated. However, several issues should be addressed to strengthen the empirical claims:

The paper is generally well-written and logically organized. The motivation and contributions are easy to follow, and the architectural components are clearly described. Some figures could benefit from better labeling (e.g., Figure 1), but the text is otherwise clear. In Figure 1, the meaning of the bars is unclear due to missing axis labels and legends. Proper labeling is essential for interpreting results.

Computational Overhead: The paper does not report the computational cost of the Persistent Event Memory Forest, in terms of runtime or memory usage. This is particularly important for streaming applications, and should be quantitatively evaluated.

---

> ### Author Rebuttal · Authors · 2025-07-28
>
> We are very grateful for your constructive comments, which have been instrumental in improving our paper. The following is a point-by-point response to the reviewer's comments.
>
>
> >**W1: In Figure 1, the meaning of the bars is unclear due to missing axis labels and legends. Proper labeling is essential for interpreting results.**
>
> Thank you for your valuable suggestion. We will revise Figure 1 in the updated version of the manuscript to ensure the results are clearly interpretable.  Specifically, we will enhance the clarity of figure by: (1) adding a clear legend indicating that the blue bars represent Previous sota, the gray bars denote StreamForest (1k visual token limit), and the green bars denote StreamForest (8k visual token limit); and (2) explicitly labeling the vertical axis to indicate that the bar heights correspond to the average accuracy on each evaluation benchmark.
>
>
> >**W2: Computational Overhead: The paper does not report the computational cost of the Persistent Event Memory Forest, in terms of runtime or memory usage. This is particularly important for streaming applications, and should be quantitatively evaluated.**
>
>
> Thank you for raising this point. We have supplemented a quantitative analysis of the runtime and memory usage associated with StreamForest. In order to better observe the impact of memory mechanism, we assume that frame-level visual features have already been extracted by the vision encoder in real time. Thus, we report the the computational cost only for the combined of LLM and memory system (PEMF + FSTW). PEMF enforces a strict upper bound on the number of visual tokens (8K in the experiments here), ensuring that memory consumption remains stable regardless of the number of processed frames. Consequently, the computational cost does not grow significantly with longer streaming inputs. The following table presents the computational resource consumption for a single effective response:
>
> |Input Frames|Memory (GB)|FLOPs (T)|Inference Latency (s)|
> |:-|:-:|:-:|:-:|
> |64|15.8|93.1|0.776|
> |256|17.1|134.1|1.126|
> |1024|17.2|137.3|1.497|
>
>
> The computational cost of updating PEMF is negligible compared to that of the vision encoder and LLM. We have added a comparison of PEMF with other memory mechanisms, including the Pyramid Memory Bank [13] and the Similarity Merge strategy [36]. The following table reports the time consumption of each method when processing a 500-second video stream (1 FPS) and producing one effective response at the 500th second.
>
> |Method|Vision Encoding (s)|Memory Update (s)|LLM Inference (s)|**Total (s)**|
> |:-|:-:|:-:|:-:|:-:|
> |Similarity Merge [36]|5.198|0.183|1.388|6.769|
> |Pyramid Memory Bank [13]|5.203|0.451|1.381|7.035|
> |**PEMF (Ours)**|5.218|**0.172**|1.394|6.784|
>
> The results show that vision encoding dominates the overall runtime, processing at 95 FPS, while PEMF’s event consolidation accounts for only 0.172 seconds for 500 frames (2900 FPS). With PEMF integrated, the preprocessing speed for the LLM’s visual inputs decreases by only 3 FPS (5.218+0.172=5.39s, from 95 to 92 FPS), which is negligible compared to the efficiency gains achieved by significantly reducing the total number of visual tokens.
>
>
>
> >**Q1: The ODV-Bench could be described in more detail. For example, "task examples" are mentioned in Figure 3(c), the paper does not clearly define a taxonomy of question types (e.g., object recognition, temporal reasoning, causal inference, prediction). I think such a taxonomy would be useful for interpreting performance in this benchmark.**
>
>
> Thank you for your valuable suggestions. In response to your concern about the *clarity of task types and subtask categorization*, we clarify that the classification criteria for the tasks and subtasks in Figure 3(a) are systematically explained in **Appendix B** (in the Supplementary Materials). Due to space constraints, we placed the detailed methodology there to preserve the coherence of the main paper.
>
> The "Task Examples" in Figure 3(c) illustrate representative scenarios from real-world driving, covering both normal and anomalous cases. The mapping between these examples and subtasks is also detailed in **Appendix B**.
>
> Briefly, we organize the benchmark tasks using a two-level hierarchical taxonomy, based on key considerations in autonomous driving. At the top level, we categorize the tasks into three broad scenarios according to the nature of the target objects:
>
> 1. Static traffic element perception
> 2. Dynamic object behavior prediction
> 3. Multi-agent interaction and risk reasoning
>
> Based on this, we further define specific subtasks using the following criteria:
>
> - Temporal cues involved (e.g., real-time perception vs. past memory vs. future prediction);
> - Practical requirements in real autonomous driving systems;
> - The cognitive skills required (perception, memory, reasoning, etc.).
>
> We believe this structured taxonomy provides researchers with a clear and multi-faceted view of model performance in video understanding tasks. We will also further strengthen the cross-referencing between the main paper and the appendix in future versions to enhance clarity and readability.
>
>
>
>
> >**Q2: Quality control in the construction of the benchmark seems under-reported. "multiple rounds of manual review," could be clarified with quantitative information regarding inter-annotator agreement, human accuracy baselines, variability as a function of taxonomy.**
>
> Thank you for emphasizing the importance of quality control in benchmark construction. To improve transparency and reproducibility, we provide the following details about our multi-stage manual review process:
>
> **1. Meta-Annotation Review**
>
> Our meta-annotations include semantic labels and spatiotemporal trajectories. Quality control strategies vary by data source:
>
> - Dynamic Target and Event-Oriented datasets: These datasets come with relatively complete raw annotations. We performed manual verification, error correction, and format standardization to generate high-quality meta-annotations.
> - Static Target datasets: For in-house or partially annotated public data, we first applied YOLO (for detection) and VLLM (for semantics) to generate initial labels, followed by multi-round manual refinement. Then, samples with major disagreement (e.g., ambiguous classes or trajectories) underwent additional review.
> - Inter-Annotator Agreement: For Static Target tasks, Cohen’s Kappa averaged 0.825, reflecting strong annotator consistency.
>
> **2. QA Pair Review Mechanism**
>
> All QA pairs went through multiple rounds of manual review, focusing on:
>
> - Relevance to the video and semantic clarity;
> - Correctness of the answer;
> - Quality and plausibility of distractors.
>
> Each QA item was independently reviewed by at least two annotators, with low-quality samples flagged and re-reviewed.
>
> **3. Human Accuracy Baselines and Task Variability**
>
> To establish human baselines, we sampled QA pairs across subtasks and conducted blind testing with trained annotators. Results are as follows:
>
> |Method|Static Target|RTP|HD|KIE|TCD|DDM|PTM|Avg.|Dynamic Target|AP|LP|DP|Avg.|Event Oriented| RP|RA|ARA|Avg.||Overall|
> |:-|:-:|:-:|:-:|:-:|:-:|:-:|:-:|:-:|:-:|:-:|:-:|:-:|:-:|:-:|:-:|:-:|:-:|:-:|:-:|-:|
> |Human|-|96.80|97.56|98.17|95.68|95.85|94.41|95.85|-|83.73|87.85|90.44|88.18|-|91.92|94.89|92.97|92.54||91.36|
>
> These results reveal varying task difficulties:
>
> - Dynamic Target tasks are most challenging (88.18%), due to their reliance on temporal cues, causal sequences, and future trend prediction
> - Event-Oriented tasks involve complex reasoning and multi-agent interactions but offer clearer context, yielding higher performance (92.54%).
> - Static Target tasks are visually grounded and relatively simple, resulting in the highest accuracy (95.85%).
>
> These findings illustrate the cognitive diversity across tasks and provide a solid reference for evaluating model performance. We will include more statistical plots and sampling details in future versions to further improve clarity.
>
> >**Q3: Joint ablations across e.g. FSTW and PEMF could be included to identify synergistic benefits.**
>
> Thank you for your valuable suggestion. We have added a joint ablation study to evaluate the synergistic benefits of the Fine-grained Spatiotemporal Window (FSTW) and Persistent Event Memory Forest (PEMF). The results are presented in the table below.
>
> |Model Variant|OVBench (Avg)|OVO-Bench (Overall)| MLVU (M-Avg)|
> |:-|:-:|:-:|:-:|
> |w/o FSTW & PEMF|58.0|52.5|51.8|
> |w/o FSTW|59.1|53.7|69.4|
> |w/o PEMF|58.9|53.5|56.6|
> |**Full Model (Ours)**|**60.5**|**55.6**|**70.0**|
>
>
> The results show that removing both modules leads to the most significant performance drop. Using either FSTW or PEMF alone improves performance compared to the baseline, but the best results are achieved when both components are integrated. This joint ablation confirms that FSTW and PEMF provide complementary benefits. FSTW enhances real-time spatiotemporal perception near the query timestamp, while PEMF supports efficient and persistent long-term memory, together yielding the strongest overall performance.
>
> >**L1: The limitations are only discussed in the SM, I think limitations should also be summarized in the main ms.**
>
> Thank you for your valuable suggestion. Due to the page limitations of the initial submission, we did not include a discussion of the method’s limitations in the main manuscript. In the revised version, we will summarize and incorporate the key limitations into the conclusion section of the main paper.

---

> > ### Comment · Reviewer_nvw1 · 2025-08-04
> > **follow-up**
> >
> > Thanks for the extensive response - my concerns are addressed and I think these ms additions will help refine the overall contribution. I'll update my scores accordingly.

---

> > > ### Author Response · Authors · 2025-08-06
> > >
> > > Dear Reviewer nvw1,
> > >
> > > Thank you very much for your constructive review and for taking the time to engage with our rebuttal. We sincerely appreciate your thoughtful feedback and are glad to hear that your concerns have been addressed. Your acknowledgment and updated evaluation are encouraging and greatly appreciated.

---

### Author Response · Authors · 2025-08-09
**Summary of the Rebuttal Stage by Authors**

We sincerely thank all reviewers for their efforts during both the review and rebuttal stages, as well as the area chair for their time and dedication in ensuring a fair and rigorous evaluation process.

We are pleased to have received encouraging feedback in the initial review stage, with ratings of 5 (nvw1), 4 (WUtt), 3 (gQt1), and 5 (AFDL). Reviewers highlighted several strengths of our work, including novel architecture (nvw1, AFDL), meaningful advance (nvw1), real practical value (nvw1), technically solid (nvw1), clearly motivated (nvw1), simple and elegant (WUtt), straightforward yet effective (AFDL), state-of-the-art performance (WUtt, gQt1), well-conducted (WUtt).

During the rebuttal stage, we provided detailed, point-by-point responses to each reviewer's concerns. All four reviewers actively participated in the discussion phase and offered positive follow-up remarks:

> Reviewer nvw1: "My concerns are addressed and I think these ms additions will help refine the overall contribution. I'll update my scores accordingly."

> Reviewer WUtt: "The rebuttal has improved my understanding of the work and my overall appreciation of the paper, and I am now leaning toward an Accept."

> Reviewer gQt1: "I think after the author's rebuttal, my concerns are mostly addressed. I will slightly increase my score to borderline accept."

> Reviewer AFDL: "Thank you for the detailed explanation. My concerns have been addressed."

We will carefully incorporate all reviewer suggestions into both the main text and the appendix, ensuring thorough and precise revisions. We are confident that the camera-ready version will meet the reviewers' expectations. We greatly appreciate the reviewers' and the area chair's dedication to a thorough and constructive review process, which has significantly contributed to improving the quality of our paper.

---

### Decision · Program_Chairs · 2025-09-17

**Decision:**

Accept (spotlight)

**Comment:**

This paper introduces StreamForest, a streaming video understanding framework for MLLMs with two key modules: (1) Persistent Event Memory Forest (PEMF) for long-horizon, event-level memory under tight token budgets; and (2) Fine-grained Spatiotemporal Window (FSTW) for high-resolution short-term perception. Empirical results are provided across multiple benchmarks to support its strong efficacy.

Overall, all reviewers view the problem as important and found the proposed approach simple and effective. Meanwhile, they flagged some major concerns: 1) the computational overhead of PEMF should be thoroughly analyzed; 2) more ablations should be provided to quantify the gains from architecture vs. data; and 3) certain parts lack clarity in writing and presentation, and need revision

The rebuttal is considered, which successfully address these concerns. As a result, all reviewers unanimously updated their scores to positively support the paper.

For the final version, please ensure that all promised revisions from the rebuttal stage are fully integrated to ensure the clarity and quality of this NeurIPS publication.